# Decision level integration of unimodal and multimodal single cell data with scTriangulate

Guangyuan Li [1,2], Baobao Song [3,4], Harinder Singh [5], V. B. Surya Prasath [1,2,6,7], H. Leighton Grimes [3,4,6] ✉ & Nathan Salomonis [1,2,4,6,7] ✉

Decisively delineating cell identities from uni- and multimodal single-cell datasets is complicated by diverse modalities, clustering methods, and reference atlases. We describe scTriangulate, a computational framework to mix-and-match multiple clustering results, modalities, associated algorithms, and resolutions to achieve an optimal solution. Rather than ensemble approaches which select the "consensus", scTriangulate picks the most stable solution through coalitional iteration. When evaluated on diverse multimodal technologies, scTriangulate outperforms alternative approaches to identify high-confidence cell-populations and modality-specific subtypes. Unlike existing integration strategies that rely on modality-specific joint embedding or geometric graphs, scTriangulate makes no assumption about the distributions of raw underlying values. As a result, this approach can solve unprecedented integration challenges, including the ability to automate reference cell-atlas construction, resolve clonal architecture within molecularly defined cell-populations and subdivide clusters to discover splicing-defined disease subtypes. scTriangulate is a flexible strategy for unified integration of single-cell or multimodal clustering solutions, from nearly unlimited sources.

Single-cell genomics has significantly evolved from its original introduction, enabling the analysis of millions of cells and many molecular modalities in complex organisms. The repertoire of simultaneous detectable modalities in a single cell continues to grow, which now includes RNA abundance, DNA accessibility, surface antigen expression, somatic mutations, and epigenetic modifications, among others[1–4]. To identify cell populations defined from multiple modalities in a single experiment, new bioinformatics approaches have begun to emerge. These include methods such as Seurat-WNN, which applies the weight-nearest neighbor (WNN) approach to integrate clustering results from two or more supported modalities[5], and TotalVI, which leverages a variational autoencoder to infer the joint latent space considering RNA and surface protein expression[6]. Notably, such methods can produce improved results over single-modality clustering methods and new gene regulatory insights. However, these and other methods are: (1) limited to specific modalities, (2) cannot jointly consider prior annotations from label transfer methods, and (3) are limited to specific types of unsupervised clustering algorithms for all modalities. Importantly, such existing integration methods inherently produce extremely variable results, dependent on the software resolution selected[7–9].

It is likely that no single-resolution in single-cell analyses will identify the most discrete cell states across different modalities, but

[1]Division of Biomedical Informatics, Cincinnati Children's Hospital Medical Center, Cincinnati, OH, USA. [2]Department of Biomedical Informatics, College of Medicine, University of Cincinnati, Cincinnati, OH 45267, USA. [3]Division of Immunobiology, Cincinnati Children's Hospital Medical Center, Cincinnati, OH, USA. [4]Immunology Graduate Program, College of Medicine, University of Cincinnati, Cincinnati, OH 45267, USA. [5]Center for Systems Immunology and the Department of Immunology, University of Pittsburgh, Pittsburgh, PA, USA. [6]Department of Pediatrics, University of Cincinnati School of Medicine, Cincinnati, OH, USA. [7]Department of Computer Science, University of Cincinnati, Cincinnati, OH 45221, USA. ✉e-mail: lee.grimes@cchmc.org; nathan.salomonis@cchmc.org

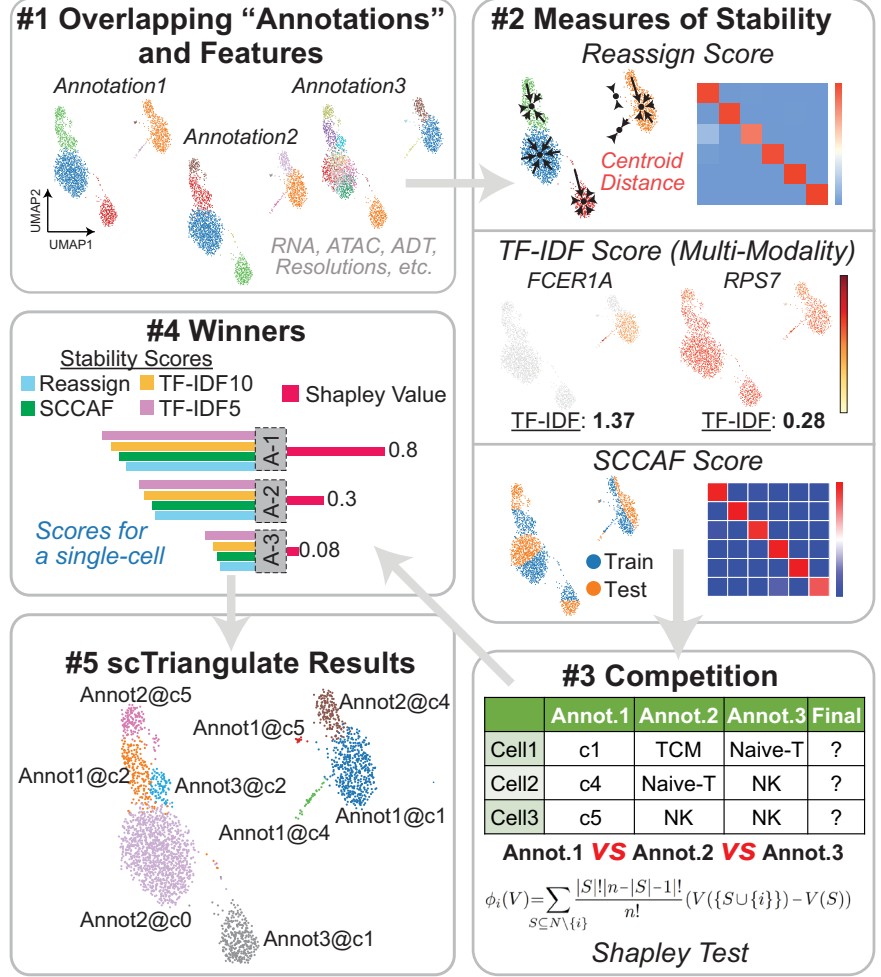

**Fig. 1 | Resolving conflicting cluster identities from decision-level integration.** Overview of scTriangulate. Using multiple competing annotation sources (algorithm, modality, and resolution), different statistical stability metrics are computed for each cluster and cell. Each cell is assigned to the winning cluster label based on a computed importance score (Shapley or Rank) (Steps 1–5, Methods). The reassign score considers the extent to which cells within a cluster can be reclassified to their own centroid based on the nearest neighbor. The TF-IDF score corresponds to the statistical strength of the $n$th-ranked feature (gene, ADT, peak) in a cluster. The SCCAF score implements a prior reliability metric (Single Cell Clustering Assessment Framework) that leverages multivariate multi-class logistic regression.

rather different informative populations (i.e., sub-clusters and clonal-specific impacts) will be highlighted when considered for a range of possible resolutions. These problems are compounded with the publication of hundreds of single-cell atlases, which pose the enormous challenge of combining and reconciling population labels from prior predictions (unimodal or multimodal). While consensus clustering through ensemble approaches, such as the Meta-CLustering Algorithm (MCLA), provides a possible solution to this problem, these methods only consider the agreement between diverse clustering results or modalities, without considering statistical measures of cluster stability or the specificity of markers in individual populations[10]. Such stability metrics, including recent robust machine-learning approaches, such as the Single Cell Clustering Assessment Framework (SCCAF) algorithm[11], are an important advance to resolve this challenge. Hence, to address these issues, there is a significant need for integrated approaches that: (a) are agnostic to the modality/algorithm/resolution considered, (b) integrate between different clustering solutions (mix-and-match), and (c) consider the biological stability of each cluster relative to all possible considered alternatives.

Here we describe scTriangulate, a conceptual framework and associated computational workflow that overcomes these challenges by using the Shapley Value adopted from game theory in combination with recent methods to compute cluster stability. While game theory

has been leveraged extensively in multiple domains, including economics, military strategy, transportation, and genetics, it has not been previously exploited in single-cell genomics[12,13]. scTriangulate systematically mixes-and-matches cellular clusters generated by different algorithms spanning different modalities and resolutions to arrive at optimal clustering solutions (based upon cluster stability) (Fig. 1). We consider each set of cluster annotations as an individual player and leverage the Shapley value from coalitional game theory to determine the relative importance of each set of cluster annotations[14]. scTriangulate considers distinct clusters produced from different unsupervised software clustering resolutions, clusters obtained from different single-cell modalities (e.g., RNA and ATAC), or annotations projected from well-defined reference cell atlases. While other multimodal methods integrate at the data level (through either a low-dimensional latent space[6] or through a geometric graph[5]), scTriangulate integrates results at a decision level to reconcile conflicting cluster-label assignments (Supplementary Fig. 1). scTriangulate is integrated seamlessly into the highly adopted scanpy python framework, enabling end-to-end analysis of multimodal and single-cell data. This flexible design makes it adaptable to new modalities (e.g., splicing and mutations), stability metrics, automated sub-clustering (e.g., subclonal analyses), and new types of research questions. Hence, this computational approach addresses numerous challenges not

addressed by existing approaches, in particular: (1) overcoming arbitrary user-defined cluster resolutions that can result in biologically incoherent populations, (2) quantitative assessment of cluster reliability (stability, doublets) across the different modalities, (3) integration across algorithm or modality-specific results, and (4) ability to support new modalities. As it is not restricted to any specific algorithms, scTriangulate players can include existing multimodal clustering algorithms (e.g., Seurat-WNN and TotalVI), which may use their own specialized batch-effect correction methods. To enable careful evaluation of its output, this tool builds a diversity of web-based interactive displays, to evaluate marker specificity, doublet predictions, cluster prediction confidence, and the contribution of specific modalities to the final solutions.

## Results

### Uni- and multimodal integration at a decision-level

A fundamental assumption of scTriangulate is that no one algorithm or modality (set of molecular features) at a given resolution can provide optimal cell clustering solutions. Such stable solutions only emerge by combining (or triangulating) the low- and high-resolution cellular views derived from complementary approaches within a decision-level integration framework, as opposed to data-level integration that seeks to obtain a joint embedding of the data. At its core decision engine, scTriangulate resolves conflicting clusters for every single cell by calculating the importance of each annotation, evaluated by a series of stability metrics. To ensure final clusters are robust, the stability of each cluster is assessed by an array of default and customizable stability metrics. Here, we assume that a valid cell population should possess a coherent molecular program (e.g., gene expression, epigenetic, and post-transcriptional) to enable accurate reclassification, with distinct detected molecular markers. Specifically, we measure whether cells in a population can be correctly reclassified against its own centroid (reassign score)[8], the exclusivity of marker abundance (e.g., gene expression, epigenetic profile, and post-transcriptional regulation) via term frequency-inverse document frequency (TF-IDF score)[15], and a combined multivariate single-cell clustering assessment framework (SCCAF score)[11]. Diverse stability metrics were intentionally selected to assess stability from distinct statistical perspectives. To assess stability across all modalities, all molecular features are combined into a single object and jointly evaluated.

While this stability ranks alone can be used to assess the validity of a cluster assignment for individual cells, the goal of scTriangulate is to assign every single cell the annotation with the highest collective rank for all stability metrics, while ensuring one algorithm does not always drive the decision. As inspiration, we considered the concept of inferred feature importance in machine learning, which is used in methods such as the SHAP value (SHapley Additive exPlanations)[16]. The Shapley Value is used to gauge the relative contribution of each annotation to guarantee fair credit allocation[14]. Rather than simply combining the stability ranks from each cell, Shapley maximizes overall cluster reliability by assessing the relative contribution of each player for all possible player combinations (coalitional iteration) for each cell. If a scTriangulate final cluster represents a small fraction of cells from the original cluster, the program automatically removes these cluster assignments (pruned, Methods) and re-classifies them against the remaining final clusters. This filter ensures that outlier cells in a cluster, which do not selectively associate with a specific user-provided annotation and are absorbed into neighboring clusters. For multi-modality datasets, features from each modality are jointly considered (e.g., RNA, ATAC, ADT, splicing, and mutation), with the contribution of each modality to each final cluster reported (Methods). Hence, while an initial cluster input in scTriangulate may have been defined by one modality (e.g., RNA), features from other modalities can support the stability of that population (e.g., ATAC). scTriangulate has no upper limit on modalities because it selects/highlights

biological features from individual modalities in obtaining the final clustering solution.

As an initial test of scTriangulate to identify the most valid cell populations from heterogeneous inputs, we created a simulated scRNA-Seq dataset using Splatter (Fig. 2a). This simulated dataset was designed to include both broad cell types and weekly distinguished cell states with unique gene expression as our ground-state truth. As the primary input to the software, we created alternative clustering solutions, in which each solution only partially matches our ground-state-truth (Fig. 2b). Although the ground-state solution was not provided to scTriangulate, it was able to reproduce this solution by mixing and matching different clusters from the broader (under-clustered) and more granular (over-clustered) inputs, without including these artifacts (Fig. 2c). Importantly, this outcome could not be obtained through tested conventional ensemble clustering algorithms, which only consider consensus and not feature-based stability (Fig. 2d, e). Additional simulation analyses demonstrate that scTriangulate can further integrate and resolve even more discrete gene expression differences by considering additional stability scores that promote weakly stable populations (Supplementary Fig. 2).

### scTriangulate finds unannotated common cell states across uni- and multimodal platforms

To determine whether scTriangulate multimodal integrated results: (1) correspond to well-described cell states, (2) have improved accuracy over alternative approaches, and (3) reveal new discrete cell populations, we first applied it to several independent human immune single-cell datasets assayed with four distinct approaches: scRNA-Seq (RNA), CITE-Seq (ADT + RNA), multiome (ATAC + RNA) and TEA-Seq (ADT + ATAC + RNA). For the analysis of snATAC-Seq, scTriangulate adopts a modified version of epiScanpy[17] to collect peak-level information for the ATAC cell clusters. For ADTs, scTriangulate uses Centered Log Ratio (CLR) normalization[1]. For these analyses, we consider a spectrum of software resolutions for each modality in the triangulation. For TEA-Seq, our initial clustering considered 9 annotation sets from all three modalities and three Leiden resolutions, resulting in 11–38 clusters per resolution that span 203 possibilities (Supplementary Fig. 3a). In contrast, projection of labels from a PBMC reference atlas (Azimuth) assigned labels for 14 cell populations, with greater than 10 cells (Fig. 3a). Using scTriangulate, we obtain 19 final cell populations composed of source clusters from all considered modalities and resolutions (Fig. 3b,c). Considering the Shapley-associated confidence of the final clusters finds that nearly all cell populations are high confidence, with unannotated scTriangulate proposed populations (not in Azimuth) resulting in lower confidence but stable predictions (Fig. 3d). Importantly, scTriangulate finds all predicted immune cell-types assigned by a comprehensive multimodal reference PBMC donor atlas[5], including challenging to detect and rare immune populations in the blood (NK CD56 + bright, B-memory and HSCP). Notably, scTriangulate finds that distinct modalities (gene accessibility, transcriptomic or cell-surface features) often distinguish highly related cell populations. For example, B-memory cells were resolved primarily by ATAC-Seq features, CD8 naive by ADT, and B-naive by RNA (Fig. 3e). In addition to these well-defined populations, scTriangulate identify subclusters not identified by Azimuth (CD4 and B-cell), frequently defined by gene accessibility or cell-surface proteins. For example, we observe the subdivision of CD4 + T-cells by the unique accessibility of a critical early T cell progenitor regulatory factor (*ZBTB17*[18]), which was not resolved at the level of gene expression (Supplementary Fig. 3b). These and other subdivisions, were further supported by cell surface protein expression in T-cells (CD45RA, CD95, and KLRG1) (Fig. 3f). To benchmark these predictions, we compared cluster assignments for individual modality clustering solutions, multimodal integration and ensemble clustering for a range of software resolutions to the Azimuth reference (Fig. 3g). Here, scTriangulate was found to have improved

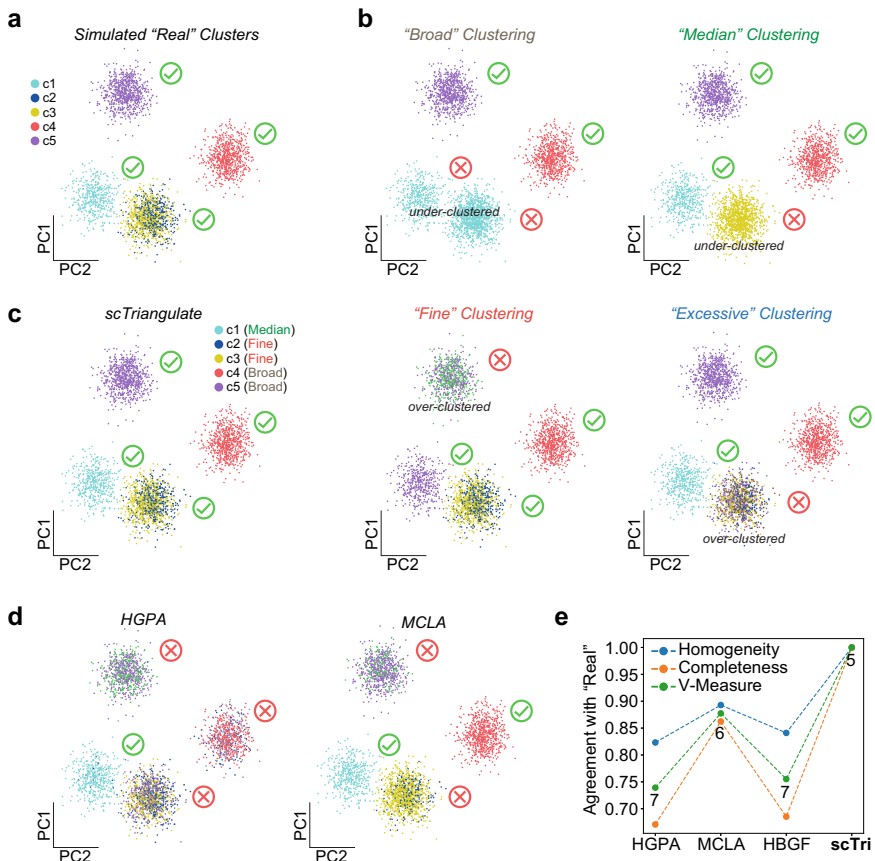

**Fig. 2 | Heterogeneous scRNA-Seq clusters are uniquely defined by scTriangulate. a** As a ground-state truth to assess the specificity of predictions by scTriangulate, we simulated scRNA-Seq data for three highly distinct simulated populations (c1, c4, and c5) and two subtly different subclusters (c2 and c3) (Methods). **b** To predict such clusters, we produce four cluster annotations for the same cells in panel (**a**), with different granular cluster definitions (broad, median, fine, and excessive). These include the noted under-clustered and over-clustered populations, which are not supported by gene expression differences. No individual resolution intentionally mirrors ground-state truth. Green checkmark = matching cluster to simulated. Red X = non-matching cluster to simulated. **c** scTriangulate results using default parameters. The "resolutions" from which each cluster was selected, are indicated in the cluster legend. **d** Example ensemble clustering outputs from HyperGraph Partitioning Algorithm (HGPA), Meta-Clustering Algorithm (MCLA), and Hybrid Bipartite Graph Formulation (HBGF) applied to the four input resolutions. **e** Benchmarking of the cluster assignments using three performance metrics (Homogeneity, Completeness, and V-Measure) for three ensemble clustering techniques and scTriangulate. The number of clusters is indicated below V-Measure. Source data are provided as a Source Data file.

performance over all evaluated solutions and resolutions, including state-of-the-art multimodal integration (Seurat-WNN and totalVI) and ensemble approaches (MCLA, HGPA, and HBGF).

To assess the reproducibility of scTriangulate on independent datasets, we generated CITE-Seq on peripheral mobilized human total nuclear cells (TNC) from two donors (donors 1 and 2). We again consider a set of clustering solutions varied across resolutions and modalities (Supplementary Figures 4a). scTriangulate successfully detected nearly all reference-defined cell populations in both samples, except for rare CD56 + high NK cells and plasmablasts in the first donor and intermediate B-cells in the second donor (Supplementary Figures 4b–g). In the case of CD56 + high NK, this population was detected in the triangulation of donors 1 and 2 (ADT-defined) but was "pruned" for donor 1, as only a fraction of the original cells was retained in the final result using default thresholds. Beyond these reference Azimuth populations, scTriangulate uncovered distinct subtypes of monocytes and MAIT cells, defined by the unique combination of RNA and ADT features (Supplementary Figures 4h–j). Importantly, these distinct subsets of MAIT cells, previously proposed to have increased responsiveness to innate cytokine stimulation, could only be derived through the integration of ADT and RNA clusters, as no source resolutions identified these populations[19]. Similar to TEA-Seq, scTriangulate of both donors again had improved performance against all individual modality resolutions (Supplementary Fig. 4k).

While these scTriangulate results show strong concordance with previous annotations, when applied to multiome RNA + ATAC PBMC this approach yields many high-confidence unannotated cell populations directly attributed to ATAC-seq features (monocytic, T cell) (Fig. 4a–c, Supplementary Fig. 5a–c). As such, the relative performance of scTriangulate by V-measure and Completeness was lower than a subset of independent clustering predictions (Supplementary Fig. 5d). However, comparison of the molecular markers in these proposed scTriangulate clusters indicates they represent: (1) classical, intermediate, and inflammatory CD14 monocytes[20] (Supplementary Fig. 5e), (2) effector memory, naïve and central memory CD4 T-cell subsets[21,22] (Fig. 4d, e), and (3) *CASC15*-defined innate lymphoid cells (ILC)[23] (Fig. 4e). In addition to being evidenced by distinct and common ATAC and RNA features, these same scTriangulate monocytic populations and associated markers were evidenced in the CITE-Seq (Supplementary Fig. 5f), demonstrating the scTriangulate predictions are highly sensitive and specific across diverse software resolutions and modalities.

To determine whether unimodal scRNA-Seq alone could identify similar populations defined by multimodal analysis, we only considered scRNA-Seq for different software settings (resolutions) (Supplementary Fig. 6a). As expected, the optimal solution defined by scTriangulate does not specifically correspond to a single individual Leiden resolution, but rather a mixture of multiple resolutions

(Supplementary Fig. 6b, c). Further, scTriangulate confidently identified 22 out of the 25 Azimuth cell populations with >10 aligned cells (Supplementary Fig. 6d, e). Notably, these "missed" populations (B intermediate, CD4 Proliferating, NK CD56 bright) were not identified

from the finest tested resolution and thus could not be considered in the triangulation. scTriangulate predictions included subtle population differences in diverse T-cell subsets, such as the correct split between Azimuth predicted CD8 T-effector memory (TEM) and CD8

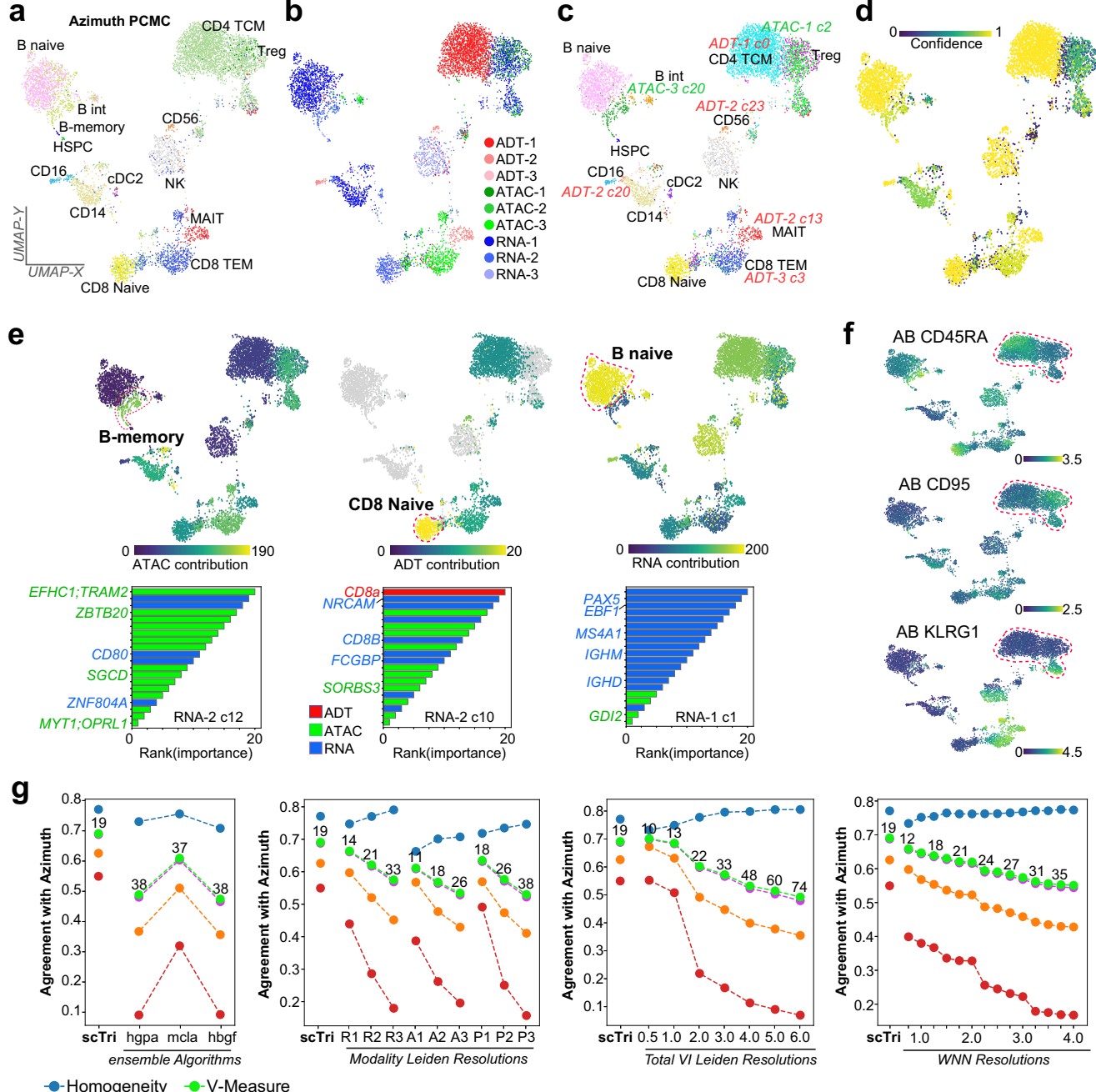

**Fig. 3 | scTriangulate identifies discrete modality-specific populations with improved performance over alternative methods. a** Supervised assignments from prior CITE-Seq cluster PBMC predictions (Azimuth). **b** Integration of nine independently obtained clustering predictions from tri-modal GEX + ADT + ATAC (TEA-Seq) in donor PBMCs. Each color indicates a distinct modality-specific clustering resolution (1-3), with the winning scTriangulate clusters shown (UMAP based on ADTs). **c** Clusters specifically derived from ADT or ATAC cluster resolutions are labeled in the plot. **d** Confidence of each scTriangulate-defined cluster. **e** The selective contribution of each indicated modality is overlaid on the UMAP (top panels), based on the frequency of associated features among the top 20 markers of each final cluster (scTriangulate visualization). In the bottom panel, the top-ranked features for representative clusters are indicated by their modality (color),

with example features denoted. f) Unannotated T-cell subsets evidenced by cell-surface ADTs. **g** Agreement with Azimuth PBMC reference label assignments measured by Homogeneity, Completeness, and V-Measure, for each individual or integrated set of clustering solutions (single-modality Leiden, multimodal Seurat WNN or TotalVI, ensemble clustering). Ensemble clustering was performed using 9 independent Leiden clustering results derived from the three independent TEA-Seq modalities (RNA, ATAC, and ADT), using three popular consensus algorithms; HyperGraph Partitioning Algorithm (HGPA), Meta-Clustering Algorithm (MCLA) and Hybrid Bipartite Graph Formulation (HBGF). The number of clusters produced by each indicated resolution is displayed above the V-Measure. Source data are provided as a Source Data file.

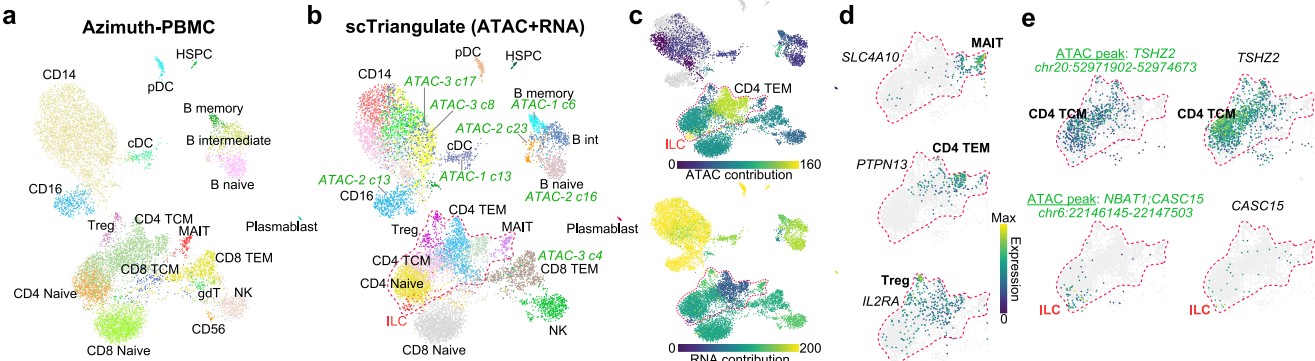

**Fig. 4 | Multimodal integration identifies rare and unannotated cell states distinguished by DNA accessibility. a** UMAP of multiome PBMC (RNA + ATAC) following Azimuth assignment of cell population identities. **b** scTriangulate final clusters after integration of multiple RNA and ATAC clustering resolutions. **c** The selective contribution of each indicated modality is overlaid on the UMAP, based on the frequency of associated features among the top 20 markers of each final cluster. **d, e** Visualization of the top-multiome RNA or ATAC markers for indicated rare or unannotated T-cell subsets. **e** Correspondence of the abundance of unique ATAC-Seq markers (e.g., *TSHZ2* marks canonical naïve/CM[43]) with RNA expression of the same localized gene.

T-central memory (TCM) cells. Such predictions result from improved stability scores and Shapley values for each evaluated metric (Reassign, SCCAF, TF-IDF) (Supplementary Figure 6f,g). Beyond these Azimuth predictions, which only find a single CD14 monocyte population, scTriangulate finds the same subdivisions of classical, intermediate, and inflammatory CD14 monocytes, as scTriangulate's CITE-Seq and multiome analyses (Supplementary Fig. 6h). Finally, to ensure the default parameters applied in this study are robust for diverse multimodal study designs, we compared them to alternative parameters, including alternative ranking strategies, cluster pruning options, and decision strategies for annotation importance for all evaluated blood datasets. These evaluations demonstrate that scTriangulate defaults result in optimal multimodal clustering integration (Supplementary Fig. 7). Hence, scTriangulate's predictions are highly accurate relative to prior references while revealing reproducible subdivisions.

### Automated cell atlas aggregation across studies

Motivated by the challenge of multimodal integration, we asked whether scTriangulate can systematically integrate and even replace cell atlas expert curation. Current protocols for cell-atlas construction require complex dataset integrations, multiple rounds of re-clustering with different resolutions and tools, alignment to different curated references and extensive manual curation by domain experts. In the lung, diverse clustering algorithms, single-cell technologies, cluster resolutions, and extensive manual curation have been used to identify and characterize dozens of distinct cell types. Indeed, projecting 142 labels from four lung atlas-level studies onto a single dataset[24] (Methods) illustrates significant diversity and conflicts in cell-type heterogeneity, cluster boundaries, and labels[25–27] (Supplementary Fig. 8a). To resolve these conflicts, we applied scTriangulate, which produced a single high-confidence and comprehensive lung atlas, borrowing the best cell-type definitions from independent studies (Fig. 5a–d). For example, scTriangulate was able to subdivide dendritic cells (DCs) and alveolar macrophages into more informative sub-populations than present in any single-reference lung atlas (Fig. 5e)[28]. Comparison of scTriangulate predictions to a cross-tissue mononuclear phagocytes atlas called MNP-VERSE, supports our finding of expanded DC and macrophage heterogeneity in the lung (Supplementary Figure 8b)[29]. While in many cases, more granular cluster predictions are rejected by scTriangulate (e.g., ncMono and cMono in Adams[26]) (Supplementary Fig. 8c), we cannot rule out the possibility that such populations will be better resolved in larger patient cohorts, by using a single-nucleus based reference, including more granular cell type predictions in the triangulation (e.g., MNP-Verse), or considering additional modalities.

Further, as the naming of populations will often vary between studies (example in Supplementary Fig. 8d), expert curation of the scTriangulate integrated labels will always be required. To further assess the quality of scTriangulate cluster predictions and names, we compared these to two independent integrated lung atlas annotation efforts (Human Lung Cell Atlas (HLCA) and CellRef, Methods) (Fig. 5f and Supplementary Fig. 8e). In addition to finding confirmatory support for many of scTriangulate's subdivisions (epithelial, endothelial, myeloid, and B-cells) and names, scTriangulate had improved agreement with independent lung atlases over those obtained with ensemble clustering (Fig. 5g and Supplementary Fig. 8f).

To determine whether scTriangulate can resolve more granular cell populations in an established cell atlas, we integrated diverse clustering annotations (ICGS2, Seurat3, Monocle3, curated) in an existing human bone marrow cell atlas composed of over 100,000 cells from eight donors[30] (Supplementary Fig. 9a–d). Importantly, the prior atlas proposed discrete hematopoietic stem and progenitor populations which have been previously described in a small subset of the original captures (<10% of cells). Importantly, these rare populations were invariably captured by these independent clustering solutions[8]. While analysis with only a single TF-IDF score failed to resolve certain transitional states (e.g., HSC-cycle, MEP, and LMPP), these were readily identified using the default options, which distinguish more granular cell states (Supplementary Fig. 9d, e). Evaluation of the default scTriangulate clusters with individual source annotations resulted in a greater agreement with the original author annotations (Supplementary Fig. 9f). In addition to transitional cell states, scTriangulate was able to identify prior literature curated dendritic, monocytic, and erythroid subsets, not defined by the original authors (Supplementary Fig. 9g–i).

### Resolving clonal impacts from genotype-aware single-cell RNA-Seq

A new and increasingly important form of multimodal single-cell genomics is the analysis of clonal genomic architecture and transcriptomic impacts. While the clonal architecture of cells from a patient can be identified with a growing number of multimodal protocols, no current integrative methods exist to determine how mutations contribute to neoplastic states. Therefore, we applied scTriangulate to single-cell gene expression with linked gene-mutation calls from a myeloproliferative neoplasm (MPN) patient[2]. Here, a small number of mutations were selectively detected from the same 10x Genomics scRNA-Seq MPN library, through sub-library amplification using the Genotyping of transcriptomes (GoT) protocol. Using GoT, the original authors identified wild-type hematopoietic cells and three

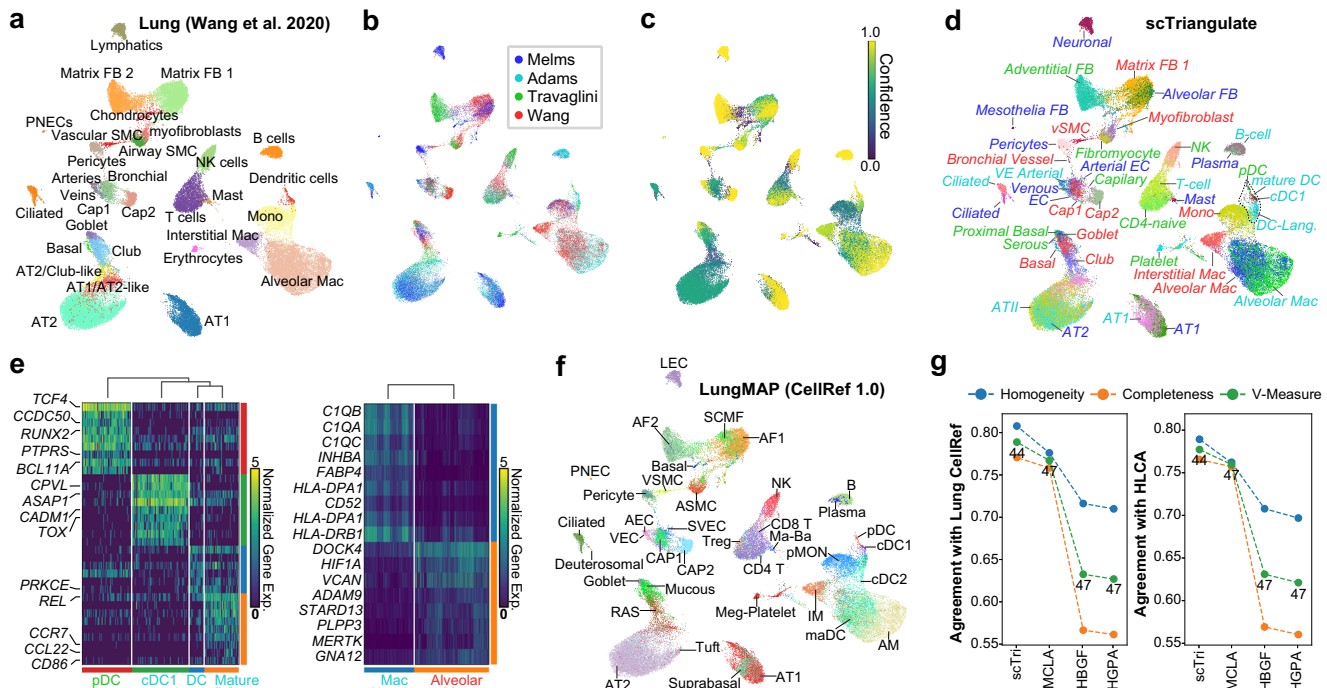

**Fig. 5 | Automated integration of cell-atlases.** scTriangulate of four competing lung cell atlases projected onto snRNA-Seq from nine donors (Wang et al. 2020)[24]. **a** Wang et al. source annotations and scRNA-Seq. **b** scTriangulate winning clusters after integrating labels from three other lung atlas studies projected onto Wang (Melms[25], Adams[26], Travaglini[27]) (Methods). **c** The confidence of each scTriangulate-defined cluster based on the "winning-fraction" of cells in each cluster (Methods). **d** scTriangulate final label assignments, corresponding to panel (**b**). **e** Heatmap displaying the top discriminating dendritic (left) and macrophage (right) cell markers for Wang et al. cluster subpopulations determined from scTriangulate (literature implicated gene symbols displayed). The cell (**d**) or label (**e**) color corresponds to the source dataset annotations in panel (**b**). **f** Cell annotations from human LungMAP CellRef labels (Azimuth projected−finest level) onto the Wang snRNA-Seq data. **g** Agreement with Azimuth CellRef (left) or HLCA (right) reference label assignments measured by Homogeneity, Completeness, and V-Measure for scTriangulate and ensemble clustering. Source data are provided as a Source Data file.

neoplastic clones (*SF3B1*, *SF3B1* + *CALR*, or *SF3B1* + *CALR* + *NFE2*) (Fig. 6a, b). First, scTriangulate integration re-assigned cells (*SF3B1* single mutants) to potentially more accurate clonal populations, as reflected by their more similar transcriptomes (Fig. 6d). Such reassignments are crucial, due to incomplete genotyping, which is inherent in methods such as GoT. Next, scTriangulate revealed *SF3B1* + *CALR* + *NFE2*-clone-induced stable cell states within most cell types sampled (e.g. HSC and MultiLin), but the *SF3B1* and *SF3B1* + *CALR* clones could only be distinguished in the MEP and early-ERP trajectory (Fig. 6c). This suggests that the genomic impact of *CALR* mutation within MEP and early ERP is unique, or that CALR impacts in earlier hematopoietic cells are eclipsed by the impact of *SF3B1*.

**Alternative splicing distinguishes subpopulations of leukemic blasts**

Similar to clonal variation in cancer, alternative splicing is an established critical driver of oncogenesis in the presence and absence of spliceosome mutations[31]. To determine whether scTriangulate, in conjunction with sparse splice-sensitive droplet scRNA-Seq, can unambiguously identify splicing-defined cellular subtypes, we profiled pediatric leukemia progenitors using the 10× Genomics 5′ platform. Since the definition of splicing-defined cell populations has not been previously successful due to the extremely sparse exon−exon junction data, we developed a strategy to identify candidate splicing-defined subclusters using gene expression-defined hematopoietic progenitor clusters (Fig. 7a, Methods). scTriangulate integration of gene expression- and splicing-defined populations identified leukemia blood progenitor populations that subdivide known cell types (Fig. 7b−e, Supplementary Figure 10a-d). For most cell types, only one splicing-defined subcluster was retained, with the

others effectively merged, based on their stability. These splicing perturbations alter the composition of numerous hematopoietic stem and progenitor (*KLF2*, *GATA2*, *GATA1*, *SPINK2*) or leukemia (*LAT2* and *NUCB2*) regulatory factors. Observed events were readily confirmed by aggregate SashimiPlot visualization (Fig. 7f and Supplementary Fig. 10e). Importantly, such clusters could not be resolved through multimodal integration of the underlying splicing and gene expression data using Seurat WNN (Supplementary Fig. 10f, g).

## Discussion

A major goal of single-cell genomics is the accurate isolation and functional characterization of predicted cell populations. Prior to validation, it is essential to have statistical confidence in the underlying predictions, with consideration of diverse possible alternatives. Given hundreds of existing approaches, rather than argue for one solution as inherently better, we argue that optimal clustering decisions require the integration of diverse cellular annotations from independent studies (i.e., reference-classification), multimodal measurements, distinct clustering algorithms, and software settings to identify the most confident, as opposed to the most common (ensemble) predictions. scTriangulate represents a fundamentally distinct approach for integration, that is fast, accurate, and can be effectively scaled to any number of new modalities or stability metrics (Supplementary Fig. 11).

As demonstrated in our evaluation studies, prior known and unannotated populations defined by scTriangulate are frequently informed by distinct modalities, can be consistently identified between different multimodal single-cell technologies, and have improved correspondence to highly curated references than prior multimodal integration approaches. Further, our results show that the integration

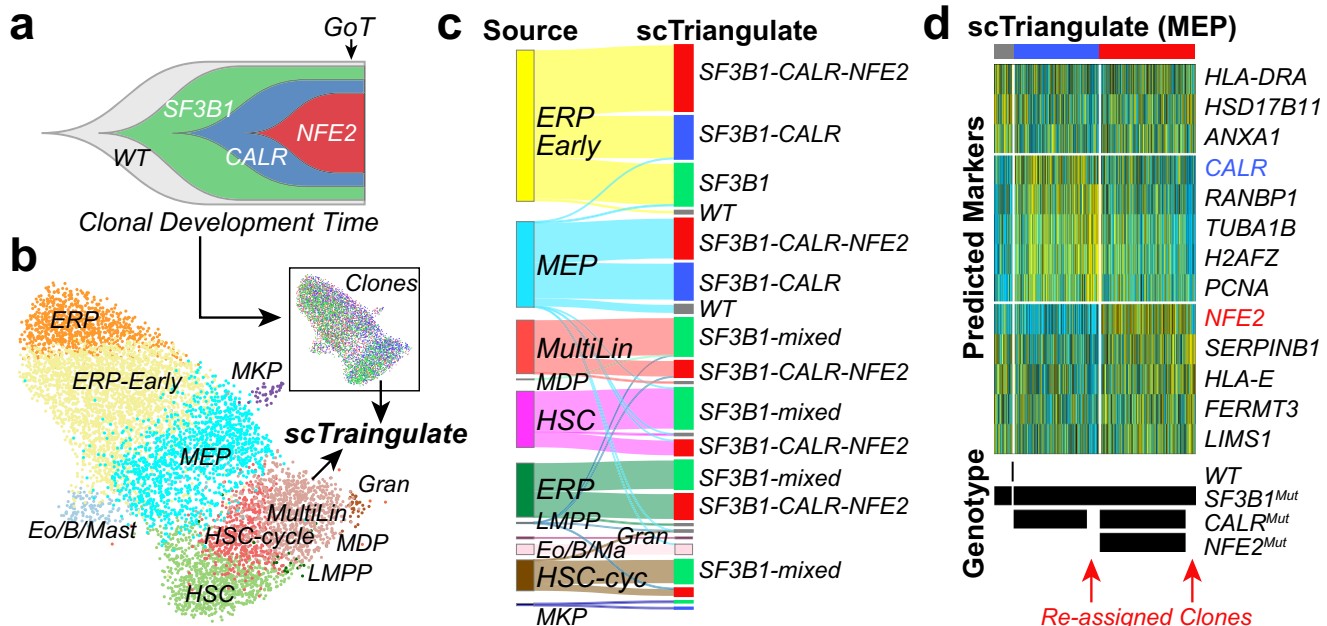

**Fig. 6 | Detection of cell-state-specific clonal impacts from genotyping-of-transcriptomes. a** Fish-plot displaying the clonal evolution of a patient with one (SF3B1), two (SF3B1-CALR), or three (SF3B1-CALR-NFE2) causal MPN mutations. GoT (arrow) indicates when single-cell profiling and genotyping were performed. **b** MPN single-cell clusters assigned by supervised assignment (cellHarmony) relative to healthy bone marrow CD34 + progenitors, with the clonal architecture of the cells indicated in the inset. **c** Sankey diagram of scTriangulate transcriptionally coherent cell populations, considering gene expression and clonal genotype. **d** Expression heatmap of final scTriangulate clusters for MEP, indicating the originally predicted genotypes from GoT and scTriangulate reassignments. Source data are provided as a Source Data file.

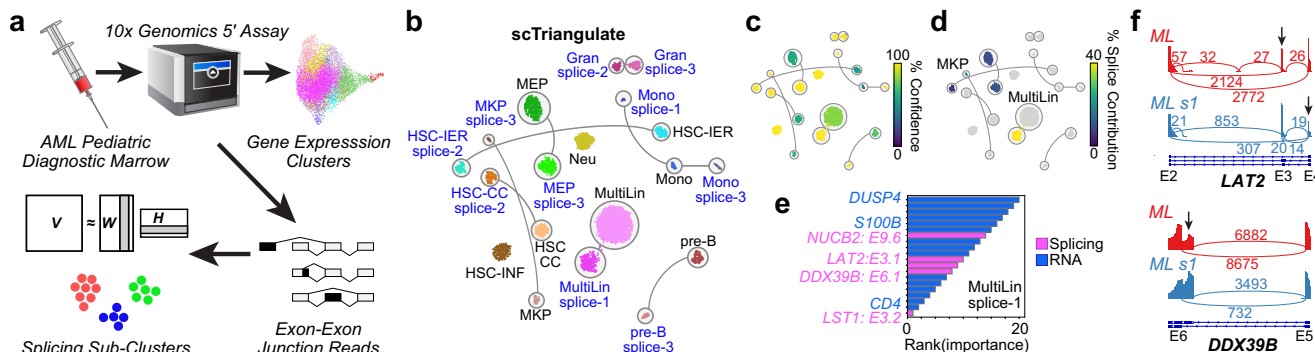

**Fig. 7 | Resolution of splicing-defined cancer cell states. a** Overview of the strategy for identifying initial splicing-defined subsets, from a diagnostic pediatric AML bone marrow sample profiled using the 10× Genomics 5′ library chemistry. Initial clusters were defined using the software ICGS2 from gene expression measurements and k = 3 sub-clusters from non-negative matrix factorization (NMF) analysis of percent-spliced-in (PSI) values. **b** UMAP of AML cells from neighborhood component analysis of AML gene expression and splicing, guided by scTriangulate analysis final "pruned" labels. The remaining stable splicing-defined sub-clusters (NMF clusters 1, 2, or 3) are indicated in blue. **c** The Shapley-associated confidence of each scTriangulate-defined cluster. **d** The selective contribution of splicing is overlaid on the UMAP. **e** Top-ranked markers (splicing and gene) in the MultiLin (ML) splicing-defined sub-cluster (s1). **f** Selected splicing event Sashimi-Plots comparing ML denoted scTriangulate sub-clusters. Source data are provided as a Source Data file.

of prior obtained annotations from lung or bone marrow identifies improved cellular identities that can only be obtained from the combination of multiple independent sources. As such, scTriangulate has the strong potential to automate cell atlas creation through the integration of different proposed annotations and new clustering results. These can include previously unidentified populations which arise from partially overlapping annotations from different sources. An important remaining challenge of such analyses is the naming of populations, which remains largely non-standardized (see Supplementary Fig. 8).

In the context of clonal analyses in cancer, we show the unique multimodal integration of gene expression and alternative splicing to illustrate splicing-defined clonal subsets in cancer as well as mutationally-defined clonal subsets that result in dominant transcriptomic impacts. While our current study was limited to the measurement of select somatic mutations within a tumor, our approach can logically be extended to the unbiased analysis of hundreds or thousands of genomic variants, identified at the DNA or RNA level using newly emerging single-cell multimodal genomics protocols[32]. Such analyses are likely to be crucial to differentiate cancer clones in vivo based on their gene, long-read, splicing, cell surface, or epigenomic readouts. Hence, as the number of single-cell modalities increases, it will be necessary to not only integrate cellular heterogeneity from each molecular readout but to combine such predictions

with reference cell annotations from existing studies that involve diverse computational tools.

While the analytical settings applied in these studies were large programmatic defaults, it is likely these settings will not be optimal for all study designs. For example, the identification of rare bone marrow progenitor populations required the use of multiple TF-IDF scores to weight the selection of rare transitional cell states with fewer unique expressed marker genes. Further, the integration of cluster predictions from dozens of sources (e.g., resolutions, modalities, reference annotations), will significantly increase runtime. For these reasons, the software includes intelligent defaults, alternative options for stability metric inclusion, and flexible parameters (e.g., pruning threshold), that allow users to customize the outputs of scTriangulate out of the box. For example, the program will automatically substitute Shapley's selection of final clusters with a combined rank-based selection, when over 15 annotation sets are provided to reduce runtime. scTriangulate can be further customized beyond such options, to include additional stability metrics or alternative rank-integration approaches.

We note that there are likely viable alternatives to Shapley-based annotation importance that should be considered in the future. When having *I* cells in total with *N* conflicting annotations for each cell, the challenge is to define a binary assignment matrix *X* of the shape *I* * *N*, where each cell can only be definitively assigned to one annotation (zero or one), such that the final clusters' stability should be maximized. Alternative solutions that could be considered include minimizing the objective function using a gradient-based solver[33], reinforcement learning[34,35], or solving as an N-player Nash equilibrium problem[36].

Looking forward, we anticipate the extension of this approach beyond single-cell genomics to more complex settings, such as spatial transcriptomics, in which the notion of molecular modalities can be extended to spatial position within a tissue, cellular niches, subcellular localization, cellular interactions, cell morphology, and other qualitative and quantitative readouts enabled through image-based analyses. scTriangulate is extendable to such approaches and already possesses an interface to the python spatial transcriptomics analysis package Squidpy[37]. In addition to the analysis of new technologies, it will be important for this and similar approaches to support the integration of multiple samples of the same modality and even across different single-cell multimodal or spatial technologies. We aim to support such features in the future, through the incorporation of innovative new computational strategies.

## Methods

This work complies with all relevant ethical regulations. The protocol for the collection and analysis of human AML peripheral blood samples was approved by the Cincinnati Children's Hospital Medical Center (CCHMC) Institutional Review Board (IRB #2018-1602). The protocol for collection and profiling of healthy peripheral mobilized donor total nuclear cells consents under the IRB-approved (#2008-1186) CCHMC Normal Donor Repository, which covers all aspects of donor recruitment, inclusion/exclusion criteria, informed consent, sample collection, processing, and storage. Consent was obtained to share the age of donor samples and the age of the range of patient pediatric samples, along with sex. Indirect patient identifiers are indicated for each sample.

### scTriangulate algorithm

scTriangulate is organized into 5 principal steps (Fig. 1). In the first step, counts or normalized data matrices are loaded in the software for QC, filtering, and scaling (optional). This step may also include running Leiden clustering sequentially through scTriangulate when no prior annotations exist. In the second step, cluster stability is computed for all input annotation sets. In the third step, an annotation importance score is computed prior to assigning each cell to a supplied or scTriangulate computed set of labels. In the fourth step, the final annotation assignments are pruned according to the fraction of final versus original cell-cluster assignment (user-defined) and re-classified using a nearest-neighbor strategy. Finally, these results are reported to the end-user using a series of interactive or static (pdf) visualizations to explore locally or over the web. Additional details and tutorials for scTriangulate are provided online with the software.

**Step 1: Annotation assignment and quality control.** Input data matrices and options are supplied to the software to enable automated processing of a dataset without existing annotations (i.e., clusters) or with externally produced annotations. This is performed using the preprocessing module wrapper function scanpy_recipe. scTriangulate begins with two data matrices **X** and **O**, where $\mathbf{X} \in \mathbf{R}^{I \times F}$ denotes the expression matrix and $\mathbf{X_{if}}$ is the scaled expression value of feature *f* in cell *i*, with the total number of features as *F* and the total number of cells as *I*. The features denote a gene, ADT, ATAC-Seq peak, or custom modality-specific features. By default, scTriangulate assumes that the input expression data is log scaled values (counts per 10,000 reads per cell + 1). Let $\mathbf{O} \in \mathbf{R}^{I \times A}$, where $\mathbf{O_{ia}}$ denotes the cluster label of cell *i* in annotation *a*, with the total number of annotations as *A*.

**Step 2: Inference of cluster stability.** To decide which cluster assignments are optimal, the software computes a series of metrics to gauge the reliability of each cluster for each annotation. scTriangulate uses four embedded scores: the reassign score, the TFIDF10 score, the TFIDF5 score, and the SCCAF score. These default scores can be augmented by users with custom metrics to suit to either promote or disqualify clusters with certain properties (doublet, cell cycle, metabolic pathway, etc). Here we describe the four default scores used in the program:

**Reassign score.** The reassign score quantifies the reliability of a cell-to-cluster assignment when compared to its own centroid (nearest neighbor distance). First, scTriangulate defines the top-ranked 30 markers using an integrative strategy considering both log-fold change (effect size) and a two-sided *t*-test *p*-value (significance) for each cluster versus all other cells. Here, the union of all discriminating features for all clusters is denoted by *L*. Next, a PCA analysis is performed on *L* to obtain a reduced dimensional representation *P*. Based on the discriminative features and the derived PCA space, scTriangulate reduces the dimension of the dataset from *F* to *P* and the centroid of each examined cluster is derived from these features, to produce a centroid matrix $\in \mathbf{R}^{P \times C}$ where *C* is the total number of clusters in each analyzed annotation set. Specifically for ranking the markers, let $r^f_{pval}(c)$ be the rank of feature *f* in cluster *c* based on the t-test p-value. The most significant feature will receive a rank=0. Then let $r^f_{lfc}(c)$ be the rank of feature *f* in cluster *c* based on the log-fold change. Likewise, the features with the largest positive log fold change will receive rank=0. The combined rank will then be:

$$r_{combined} = (r_{pval} + r_{lfc})/2 \qquad (1)$$

We define the function NC(·) to find the cluster label of the nearest centroid of each cell *i*, given the centroid matrix we defined above, and the Reassign score for each cluster *c* as:

$$S^a_{reassign}(c) = \frac{\Sigma_{i \in c}(NC(i) = c)}{|c|} \qquad (2)$$

Where |*c*| denotes the size of cluster *c*, the reassign score effectively measures the fraction of cells in each cluster that can be reclassified to its own centroid. A higher value indicates greater stability.

**TFIDF(n) Score.** The TF-IDF score measures the number of exclusively expressed features in a cluster. This score is based on the observation that genes or other features that are uniquely expressed in one cluster typically demarcate the boundary of a potentially unannotated cell population. Unlike the reassign score, which utilizes the expression values, the TF-IDF score focuses on expression frequency (zero and nonzero) as a binary value. To calculate the TFIDF score for a cluster, we first define the Term Frequency (TF) and Inverse Document Frequency (IDF) of each feature $f$ in each cluster $c$. A pseudo value $\epsilon = 1e - 5$ is added to avoid the numerical issues:

$$TF_f{}^a(c) = \frac{\Sigma_{i \in c} Indicator(X_{if} > 0)}{|c|} + \epsilon \tag{3}$$

$$IDF_f{}^a(c) = -\log(\frac{\Sigma_{i \in I} Indicator(X_{if} > 0)}{|I|} + \epsilon) \tag{4}$$

Where $I$ denote cells in the whole dataset but $c$ only contains cells in the selected cluster. Then we define the exclusivity, which is equal to the TF-IDF score of a feature in the cluster $c$ as:

$$Exclusivity_f{}^a(c) = TFIDF_f{}^a(c) = TF_f{}^a(c) \times IDF_f{}^a(c) \tag{5}$$

Next, this function ranks the features in each cluster based on their exclusivity in descending order. We can define the TFIDF($n$) score of a cluster as the exclusivity value of the **n**th feature:

$$S^a{}_{TFIDF(n)}(c) = Exclusivity_f{}^a(c)^n \tag{6}$$

We consider TFIDF5 and TFIDF10 scores by default. As an example, $S_{TFIDF(5)}(c1)$ means the TF-IDF score of the top 5th exclusively expressed feature in cluster c1. If a cluster expresses few unique marker genes (e.g., $n = 3$) but is considered biologically valid, the user can add or replace the existing TF-IDF score with an alternative rank, such as TF-IDF1 to prioritize such features in the downstream Shapley analysis (Supplementary Fig. 2).

**SCCAF score.** We reimplemented the statistical tests within the previously described Single Cell Clustering Assessment Framework (SCCAF)[11] to serve as an added metric to gauge the biological reliability of the cluster. Different from the reassign score, where we leverage the univariate markers from a two-sided t-test, SCCAF considers the marker definition challenge as a multivariate multi-class logistic regression problem, which inherently takes into account the interactions and dependencies between different features. While only the most discriminative features are considered for the reassign score, SCCAF considers all features. As it considers all features, SCCAF scores will not always be concordant with reassign and will differentially assess stability. In the SCCAF model, the expression matrix and the Annotation (cluster label) serve as the data **X** and the label matrix **O**, respectively. After the logistic regression model inference, a confusion matrix $\Psi \in \mathbf{R}^{C \times C}$ will be produced. This matrix stores the number of times the model correctly and incorrectly predicts each instance's cluster label. Using the confusion matrix, we define the SCCAF score of a cluster $c$ as:

$$S^a_{SCCAF}(c) = \frac{\Psi_{cc}}{\Sigma_{c \in C} \Psi_{cc}} \tag{7}$$

**Step 3: Shapley value determination and cell-cluster assignment.** To define the most stable cell populations from multiple alternative annotation sources, scTriangulate considers the marginal contribution of each annotation through a decision-making strategy. By default, this strategy employs the Shapley value adopted from the coalitional game theory. Similar to its original intended application, the Shapley value will quantify the marginal importance of each annotation (player)

versus others by considering the rankings produced for different stability metrics. The rank is determined considering all players within each possible coalition (all player permutations), hence, the rank of a stability metric is dependent on which players are considered in which coalition. Here, similar scores (within a tolerance threshold—aka offset) will be assigned the same rank, where ranks themselves are scaled in a "winner takes all" manner. This strategy has numerous conceptual and mathematical advantages over alternative strategies. For example, while simple rank-based annotation importance can be applied to the derived stability ranks (without coalitional iteration), fairly minor overall differences in the scores from one stability metric can override much larger differences in another and frequent ties between players. When different cluster annotations are considered equivalent (a tie), the software selects the smaller source annotation as the winner.

To compute, following the determination of stability metrics in Step 2 for every single cell, a resulting matrix $\mathbf{D} \in \mathbf{R}^{A \times M}$ will be produced, where $\mathbf{D_{am}}$ denotes the calculated metrics of an annotation an under metrics m. For instance, if one selected cell has three conflicting annotations, namely label-1, label-2, and label-3. By default, matrix D will be of the shape $3 \times 4$ as scTriangulate computes 4 scores.

We consider each annotation here as one "player" in the analysis. To assess the relative importance of a player, the Shapley Value $\Phi$ has been proposed to guarantee fair credit allocation, which considers the amount of added contribution (marginal contribution) from each player (one label) to the collective performance of a team/coalition (multiple labels). With a proper Value function V(T) to represent the team's performance, the Shapley Value of a label **a** can be computed as:

$$\Phi_a = \sum_{T \subseteq A\{a\}} \frac{|T|! \times |A - |T| - 1|!}{|A|!} (V(T \cup \{a\}) - V(T)) \tag{8}$$

In brief, the Shapley Value iterates every possible formation of the teams and measures the marginal contribution of each player (label) to the team, normalized by the team size. scTriangulate defines the added performance (surplus) as below:

**Algorithm 1.** scTriangulate Computing ($V(T \cup \{a\}) - V(T)$)
 1: Input: annotation a and metric matrix $D$
 2: V = 0
 3: **for** $m \leftarrow \{Metric1, Metric2, Metric3,...\}$ **do**
 4: **if** Rank($D_{am}$ + 0.01) = $|T \cup \{a\}|$ **then**
 5: $V \leftarrow V + |T \cup \{a\}|$

In the value function (Algorithm 1), 0.01 equals the default tolerance threshold or offset, to assign marginally close scores the same rank. The rationale for an offset is that two or more players will often be assigned a stability score for a given metric that are almost numerically equivalent but that results in different rankings. The use of an offset is inspired by General Linear Models (GLM), where an offset term is used to account for residual variation that is disproportionate and cannot be explained by the regression model itself. A fixed value offset is used as stability metric scores are in the range of 0-2 (adjustments may be required for new scores outside this range). We note that the optional use of an offset can improve overall performance in some but not all real-world benchmarking datasets (Supplementary Figure 7b). When computing the rank of a stability metric within a specific coalition, the final ranks undergo an optional adjustment. Specifically, the two existing adjustment options are "classic" and "all_or_none" (default). By default, the program re-scales all ranks for each cell and for each stability metric in a winner takes all manner (all_or_none), where only the top rank choice(s) retain its rank (all others set to 0). If multiple annotations have the top rank, then each of these annotations will be assigned the same non-zero rank. Hence, the all_or_none approach rank normalization approach only rewards a player if that cluster annotation has a higher score than all others for a given stability metric. The rationale for this option is that the best

performers will rise to the top, producing a more definitive ranking for downstream steps. Conversely, "classic" will retain all of the original ranks, post-offset consideration. The impact of these adjustment choices will impact the final results (Supplementary Figure 7c), most significantly, when simply ranking as opposed to Shapley is applied. After considering offset, the highest numerical value in $D_{am}$ will receive the highest rank (scipy convention).

scTriangulate uses the $\Phi_{a(i)}$ to denote the relative importance of each label for each single cell $i$, and assigns this single cell $i$ to the label with highest $\Phi$:

$$argmax_{a \in A} \Phi_a(i) \quad\quad (9)$$

Since the computational complexity of Shapley increases exponentially as a result of exhaustive iteration of all coalitions, compute time becomes intractable with greater than 15 annotations. Hence, to reduce runtime, the program will default to an alternative solution purely based on rank, as a reasonable alternative to Shapley when the number of user-supplied conflicting annotations is greater than 15. Evaluation of pure ranking as the method of annotation importance demonstrates that it has generally equivalent importance (depending on the benchmarking dataset and ranking strategy) when explicit ranking (classic mode) is used in place of all-or-none (winner takes all) rank adjustment (Supplementary Fig. 7c).

To assess the overall stability and quality of Shapley-selected annotations, the software includes the option to produce an average cluster Shapley value, analogous to the use of an Elo rating in chess. This quality metric is defined as the average of all cells' Shapley value in all clusters, normalized by the number of players, and further normalized by the number of clusters. Since the Shapley calculation is an additive process, the more players considered, the higher the Shapley value will be. This quality metric can be used to approximate which cluster is more stable, even for clusters derived from different datasets.

**Step 4: Results pruning and cell reclassification.** As scTriangulate assigns stable populations at the level of individual cells, an optional pruning step was introduced to exclude cell populations with few cells retained relative to the source cell-to-cluster assignments. Here, we assume that if a large initial cluster of cells exists, but only a small minority of cells with that annotation are considered to be "stable", the overall predictions are more likely to be unreliable. Such stable minor subsets will often be due to technical artifacts (e.g., high mitochondrial expression, dying cells), but may represent a valid "activation" state of a parent cell type.

For this analysis, we denote the new cell label based on the highest Shapley Value as $A^{raw}$. We further filtered out unstable clusters based on two criteria:

(1) winning fraction: $wf_c = \frac{|A_c^{raw}|}{|A_c^{ori}|}$. Stable clusters have a higher winning fraction of cells relative to the number of cells in the original cluster as compared to unstable clusters, which may win out purely by chance. By default, $wf_c < 0.25$ will be labeled as unstable clusters. The winning fraction of raw clusters indicates the confidence that the program possesses for this prediction. In a benchmarking analysis of pruning thresholds ranging from 0-60%, we find that more pruning is predicted to always improve performance, but with minimal improvement beyond a threshold of 40% (Supplementary Figure 7d).

(2) By default, $|A_c^{raw}| < 10$ will be labeled as unstable clusters. Specifically, scTriangulate only considers cell populations with a minimum number of cells (user-defined) for consideration of stability, since stability estimates in such populations are inherently less reliable due to potential outlier effects (abs_thresh parameter). Considering different possible thresholds for minimum cluster size (1-50 cells), the default threshold of 10 produced consistent reliable results for

different evaluated datasets (Supplementary Figure 7a). All cells within the unstable clusters will be reassigned to their nearest neighbor cluster label.

**Contributions of modality.** The contribution of each modality {RNA, ADT, ATAC} in each cluster c is defined as a weighted sum of its markers (gene for RNA, antibody for ADT, peak for ATAC) in the top-20 marker list for the cluster $c$, the weight is the rank of the marker in the list such that the top-1 marker will receive importance as 20. Specifically,

$$Contributionn_{RNA}(c) = \sum_{i \in top20} Indicator(i = gene) \times Rank \quad (10)$$

$$Contributionn_{ADT}(c) = \sum_{i \in top20} Indicator(i = antibody) \times Rank \quad (11)$$

$$Contributionn_{ATAC}(c) = \sum_{i \in top20} Indicator(i = peak) \times Rank \quad (12)$$

where $i$ refers to each marker in the top-20 list, which can be either a gene, antibody or ATAC peak. Rank is the rank importance defined above.

**Step 5: Results visualization.** scTriangulate results can be exported en masse to an HTML archive to interactively explore marker expression, cell population boundaries, doublet cell predictions, quality control metrics, stability metrics, Shapley Values, modality contributions, and cell-population prediction confidence. These plots can be viewed as scatter plots (e.g., UMAP), heatmap, or bar charts. UMAP plots can optionally be exported from a supervised PCA analysis, considering the final scTriangulate clustering labels in the projection, using neighborhood component analysis (NCA). In addition, these plots can be individually exported from the software. Such visualization is an essential step to guide the optimization of scTriangulate results to use alternative stability strategies (e.g., TFIDF5) or alter the cutoffs for pruning and reclassification in the stored scTriangulate object.

**scTriangulate performance evaluation**
To assess scTriangulate's ability to produce optimal uni-modal or multimodal clustering solutions, we compared its results to those to (1) highly used multimodal integration approaches, (2) ensemble clustering, and (3) individual clustering algorithms. As benchmarks, we considered prior silver-standard curated atlas annotations (e.g., Azimuth) and synthetic scRNA-Seq data produced with Splatter. While the silver standard datasets are not definitive solutions, they represent significant annotation efforts where the original authors extensively analyzed different clustering solutions (algorithms, resolutions) and/or multimodal measurements (CITE-Seq).

**Performance metrics.** To assess relative agreement, we used three complementary cluster evaluation metrics, namely Homogeneity, Completeness, and V-Measure. These three metrics were verified using Adjusted Mutual Information (ARI) and Adjusted Rand Index (ARI). Intuitively, Homogeneity denotes the extent to which each scTriangulate cluster only contains members of a single silver-standard reference cluster, or in other words, how homogeneous each scTriangulate cluster is. For annotations derived from low versus high clustering resolutions, it is expected that higher resolutions will increase homogeneity. Homogeneity is formulated as:

$$homogeneity = 1 - \frac{H(C|K)}{H(C)} \quad (13)$$

Where **H(C)** is the entropy of the silver-standard reference cell type **C**, and **H(C | K)** is the conditional entropy of the reference cell type **C** given the scTriangulate (or alternative clustering solution) cluster assignment **K**. Below, $n$ is the total number of cells in the tested single-cell dataset, $n_c$ is the number of cells for each reference class $c$, $n_{c,k}$ is the number of cells that reference class $c$ gets assigned to scTriangulate cluster $k$. Here, the mathematical notation is consistent with conventions applied in the sklearn python library.

$$H(C) = -\sum_{c=1}^{|C|} \frac{n_c}{n} \times \log(\frac{n_c}{n}) \qquad (14)$$

$$H(C|K) = -\sum_{k=1}^{|K|}\sum_{c=1}^{|C|} \frac{n_{c,k}}{n} \times \log(\frac{n_{c,k}}{n_k}) \qquad (15)$$

Conversely, Completeness aims to capture the extent to which all cells from each individual reference class are assigned to one single scTriangulate cluster. As expected, broader clusters derived from low-resolution clustering will result in improved Completeness, as the chance of fully encapsulating a defined cell type is higher, unlike Homogeneity. Completeness is defined by:

$$completeness = 1 - \frac{H(K|C)}{H(K)} \qquad (16)$$

Where the **H(K | C)** denotes the conditional entropy of scTriangulate cluster assignment **K** given the reference cell type class **C**, $n_k$ denotes the number of cells contained in each scTriangulate cluster $k$, $n_{k,c}$ denotes the number of cells that scTriangulate cluster $k$ comes from the reference cell type class $c$.

$$H(K|C) = -\sum_{c=1}^{|C|}\sum_{k=1}^{|K|} \frac{n_{k,c}}{n} \times \log(\frac{n_{k,c}}{n_c}) \qquad (17)$$

$$H(K) = -\sum_{k=1}^{|K|} \frac{n_k}{n} \times \log(\frac{n_k}{n}) \qquad (18)$$

Clustering solutions in agreement with the reference should maximize the Homogeneity Completeness score. V-measure is defined as the harmonic mean of both of these scores, which is formulated by Rosenberg and Hirschberg:[38]

$$V\,measure = 2 \cdot \frac{h \times c}{h + c} \qquad (19)$$

These metrics overcome potential pitfalls of alternatives, including no assumption of cluster structure and increased interpretability to diagnose whether a clustering resolution is too broad or too granular, which makes it suitable for evaluating single-cell clustering results, as previously demonstrated[39,40]. While informative, as silver standards are likely not to be comprehensive and often do not consider stability, results discordant with these assessments require stability, and functional and/or literature verification to assess significance.

**Simulation datasets.** scRNA-Seq simulation datasets were generated using Splatter version 1.16.1 with the *splatSimulateGroup* function. To assess the ability of scTriangulate to resolve different stable populations from different source annotations, a dataset was simulated with 3000 cells (*batchCells* = 3000). This ground-state truth dataset had 5 clusters (c1, c2, c3, c4, c5), where their size was determined by the parameter *group.prob* (0.23,0.15,0.15,0.23,0.24), in which each numerical value corresponds to the size of each simulated cluster. The extent to which two clusters are different is determined by parameter

*de.prob* (0.15,0.15,0.15,0.2,0.4), where each numerical value corresponds to the fraction of genes that are differentially expressed in each cluster. We constructed four different annotations, which only partially overlap with the ground state solution. Specifically, we produced a "Broad" annotation that merges the above c1 + c2 + c3 (under-clustered), a "Median" annotation which merges c2 + c3 (under-clustered), a "Fine" annotation that arbitrarily sub-divides c5 (over-clusters), and an "Excessive" annotation which arbitrarily sub-divides both c2 and c3. We apply the scTriangulate *lazy_run* function in version 0.9.0 on these four conflicting annotation solutions to derive the final cluster assignment (default settings).

To assess the impact of using one versus two TF-IDF scores, *batchCells* = 3000 was generated for 6 ground-state truth clusters (c1, c2, c3, c4, c5, c6) with their size determined by parameter group.prob (0.2,0.2,0.15,0.15,0.15,0.15). The extent to which two cluster groups are different is determined by *de.prob* (0.2,0.2,0.005,0.005,0.005,0.005), here we intentionally set the difference between c3-c6 to be subtle in order to assess the usage of multiple TF-IDF stability scores. Next, we constructed three different annotations, in which the "Broad" merged c3 + c4 + c5 + c6 (under-clustered), the "Median" annotation combined c3 + c4 and c5 + c6 (under-clustered), and a "Fine" annotation which successfully identify the 6 simulated populations. Here, we applied scTriangulate *lazy_run* function in version 0.9.0 using one TF-IDF (*add_metrics* = {}) and two TF-IDF scores (*add_metrics* = {'tfidf5'}) on these three conflicting annotations to derive the final cluster assignment. The PCA plot was generated using *logNormCounts* and the *runPCA* function provided in the Splatter package.

**Performance benchmarking**
We conducted three compute performance benchmarks to assess the contribution of increasing number of cells, features, and annotation sets (e.g., clustering solutions) on overall run time. The impact of varying number of cells was tested using Wang et al. 2020[24], which has 46,500 cells and 4 different annotations on randomly sampled 10,000, 20,000, 30,000, and 40,000 cells using the scTriangulate *lazy_run* wrapper function (*assess_pruned* = False, *viewer_cluser* = False, *viewer_heterogeneity* = False). The impact of a varying the number of features was tested in the multiome PBMC (GEX + ATAC) dataset using 3 annotations on all 10,991 cells, downsampled to 40,000, 60,000, 80,000, and 100,000 features and the above test function. The impact of a varying number of annotation sets was tested using the total nucleated cells CITE-Seq (donor 1), using 3, 6, 9, and 12 annotation sets of varying Leiden resolutions. Runtime was evaluated using the default multi-core setting and compared with the actual CPU time if no parallelization was applied.

**Preprocessing and quality control**
**CITE-seq total nucleated cell.** De-identified human granulocyte colony-stimulating factor (G-CSF) mobilized total nucleated cells (TNCs) were obtained through apheresis from Cell Processing Core (CPC) at Cincinnati Children's Hospital Medical Center (CCHMC). Two young adult donors were separately profiled. Donor 1 (ND19-341) was obtained from a 28-year-old white male, and donor 2 (ND19-446) was from a 31-year-old white female. For each donor, 100,000 cells were stained with 31 human TotalSeq-A antibodies (BioLegend®, San Diego, CA) cocktail for 30 min on ice, and subsequently washed with flow buffer (1% FBS in PBS) and counted. All antibodies were titrated on both TNCs and CD34 + cells at 5 serially diluted concentrations (1:25, 1:50, 1:100, 1:200, 1:400). Staining index was calculated to guide the selection of the appropriate volume of antibodies to use (Supplementary Table 1). -15,000 cells were loaded on a 10x Chromium controller with 3' GEX version 3 chemistry. The GEX libraries were sequenced with a target depth of > 50,000 reads/cell with PE150 on the Illumina S4 flow cell. ADT library was sequenced with a target depth of >10,000 reads/cell with PE50 on Illumina SP flow cell.

Raw RNA-Seq FASTQ files were aligned with Cell Ranger 3.1.0 to the human reference genome (hg19) to produce a filtered counts matrix. Cell barcode filtering, normalization, and PBMC Azimuth (level-2(L2)) reference classification were performed as with the scRNA-Seq, to yield 6,406 (replicate 1) and 4,333 (replicate 2) cells. For the ADT count, we conducted Center log-ratio (CLR) normalization using *scTriangulate.preprocessing.Normalization.CLR_normalization* function. We then conducted hypervariable gene selection (*top = 3000*), neighbor graph construction (*n_neighbor = 15*), dimension reduction (*n_pc = 50*), and Leiden clustering (r = 1,2,3) for RNA. In parallel, we performed neighbor graph construction (*n_neighbor = 15*), dimension reduction (*n_pc = 15*), and Leiden clustering (r = 1,2,3) for the ADT data. Finally, we concatenate both properly normalized RNA and ADT values to a combined AnnData through *scTriangulate.preprocessing.concat_rna_and_other* function. The combined AnnData serves as the input for the downstream scTriangulate analysis (*win_fraction_cutoff = 0.25, add_metrics = None*). All the preprocessing and QC steps were conducted using scTriangulate version 0.9.0 and scanpy version 1.7.2. The Seurat WNN analysis was performed following the official vignette (https://satijalab.org/seurat/articles/weighted_nearest_neighbor_analysis.html). Specifically, we chose 50 Principal Components (PCs) for RNA and 18 PCs for ADT, respectively. We adjusted the resolution arguments in the FindClusters function in Seurat Version 4.0 to achieve different clustering results.

**Single-cell RNA-seq of pediatric AML CD34 + progenitors.** CD34 + positive flow cytometry sorted progenitors were isolated from whole blood from a pediatric AML patient (age > 1 and < 18 years old) resistant to chemotherapy at the time of diagnosis (donor ID = CF001, sample ID = CF001CD34D0). The protocol for collection and analysis was approved by the Cincinnati Children's Hospital Medical Center Institutional Review Board. ~15,000 cells were loaded on a 10x Chromium controller with 5' GEX version 1 chemistry. The GEX libraries were sequenced with a target depth of > 80,000 reads/cell with PE150 on Illumina S4 flow cell (884,687,848 total reads). Raw RNA-Seq FASTQ files were aligned with Cell Ranger 2.1.1 to the human reference genome (hg19) to produce a filtered counts matrix. Distinct cell populations were identified and annotated using the software ICGS2 in AltAnalyze, using default parameters. Exon-exon junction counts for each cell barcode were obtained using the python library Pysam and imported into AltAnalyze for percent spliced-in (PSI) quantification using the MultiPath-PSI algorithm. Out of 346,972 detected exon-exon junctions associated with defined genes (Ensembl 72), 17,519 variable splicing events with PSI estimates were reported using default options in AltAnalyze. To identify informative splicing events that are enriched in a cell population and potentially further subdivide existing clusters, we calculated pairwise comparisons for splicing events with PSI values detected in ≥25 cells, for each cellHarmony assigned cluster versus the largest cluster (MultiLin), first ignoring missing values (not detected in the cell) and then imputing missing values from the median measurement of each event across all cells in AltAnalyze. We retained splicing events with a δPSI > 0.1 and empirical Bayes *t*-test *p* < 0.05 (limma default), FDR corrected, with matching significant comparison prediction in both the missing-value and imputed analysis for downstream sub-clustering (n = 418 events). These events were augmented with those consistently identified with the algorithm MarkerFinder using missing value PSI and imputed (*n = 66*). To define candidate sub-clusters, we performed non-negative matrix factorization in the software ICGS2, to obtain 3 sub-clusters for each cellHarmony defined cell-population (*k = 3*) as input for scTriangulate. For scTriangulate, we concatenate both properly normalized RNA and pre-imputed PSI values to a combined AnnData through *scTriangulate.preprocessing.concat_rna_and_other* function. The combined

AnnData serves as the input for the downstream scTriangulate analysis (*win_fraction_cutoff = 0.25*). The Seurat WNN analysis was performed following the official vignette (https://satijalab.org/seurat/articles/weighted_nearest_neighbor_analysis.html). Specifically, we chose 50 Principal Components for RNA and 30 PCs for PSI, respectively. We adjusted the resolution arguments in the FindClusters function in Seurat Version 4.0 to achieve different clustering results.

**Single-cell RNA-Seq and Genotyping of CD34 + MPN Progenitors.** Pre-processed count matrices for gene expression and genotyping of transcriptomes were downloaded from GEO (GSE117825). For each mutation, both wild-type and mutant allele genotypes were used to assign each cell to one of four clonal genotypes (WT, SF3B1 mutant, SF3B1-CALR mutant, or SF3B1-CALR-NFE2 mutant). These mutation assignments include both false negatives (shallow sequencing) and false positives (ambient RNAs). Cell barcode filtering and normalization were performed in the software ICGS2 with default parameters, and cells assigned to Hay et al. defined CD34 + cell populations using the software cellHarmony with default options. For scTriangulate, supplied clusters consist of those from cellHarmony and subclusters for cells from the assigned clonal genotypes. First, we concatenate both properly normalized RNA and genotype assignments to a combined AnnData object through the *scTriangulate.preprocessing.concat_rna_and_other* function. The combined AnnData serves as the input for the downstream scTriangulate analysis (*win_fraction_cutoff = 0.25*).

**scRNA-Seq PBMC.** Pre-processed sparse-matrix counts (h5) were obtained from a human 10x Chromium (3' version 3 chemistry) PBMC sample, available from the 10x Genomics website (https://support.10xgenomics.com/single-cell-gene-expression/datasets/3.0.0/pbmc_10k_v3). Cell barcodes were filtered based on a min_gene > 300, min_count > 500, pct_counts_mt < 20%, to yield 11,022 single cells. The raw UMI count matrix was adjusted to logarithmic Counts Per Ten Thousand (CPTT), we then performed hypervariable gene selection (top 3000), neighbor graph construction (*n_neighbor = 15*), dimension reduction (*n_pc = 50*), and Leiden clustering sequentially through the scTriangulate preprocessing module wrapper function scanpy_recipe. Separate Leiden clusters (annotations) were obtained using a resolution of 1,2 and 3. All the preprocessing and Quality Control (QC) were run in scTriangulate version 0.9.0 and scanpy version 1.7.2. The AnnData after Leiden clustering was subjected to scTriangulate (*win_fraction_cutoff = 0.25, add_metrics = 'tfidf5'*) as input. Azimuth L2 annotations were derived from the filtered raw count data through the Azimuth PBMC online R-Shiny app (https://app.azimuth.hubmapconsortium.org/app/human-pbmc).

**Multiome PBMC.** Multiome PBMC sparse matrix counts h5 data was downloaded from the 10× Genomics website (https://www.10xgenomics.com/resources/datasets/pbmc-from-a-healthy-donor-granulocytes-removed-through-cell-sorting-10-k-1-standard-2-0-0). We conducted QC based on both RNA and ATAC peaks. We filtered out nuclei with min_genes < 300, min_counts < 500, pct_counts_mt > 20% for RNA data, together with the additional criteria for at least 1000 peaks/nucleus in the ATAC data based on episcanpy[17] tutorial. Taken together, 10,991 nuclei were kept for further analysis. We normalized both RNA and ATAC count data to logarithmic CPTT, then performed hypervariable gene selection (top=3000), neighbor graph construction (*n_neighbor = 15*), dimension reduction (*n_pc = 15*), and Leiden clustering (r = 1,2,3) for the RNA expression. For the ATAC peaks, we performed hypervariable peak selection (top=100,000), neighbor graph construction (*n_neighbor = 15*), dimension reduction (n_pc=100), and Leiden clustering (r = 1,2,3). The arguments for n_top_features and n_pc were performed as indicated in the episcanpy

tutorial. Finally, we concatenate both properly processed RNA and ATAC values to a combined AnnData through *scTriangulate.preprocessing.concat_rna_and_other* function. The combined AnnData serves as the input for the downstream scTriangulate analysis (*win_fraction_cutoff = 0.35, add_metrics = 'tfidf5'*). The inclusion of the secondary TF-IDF score was based on a preliminary evaluation with both the default and additional tfidf5 parameters, which better reflected known cell population marker diversity. All the preprocessing and QC steps were conducted using scTriangulate version 0.9.0 and scanpy version 1.7.2.

**TEA-Seq PBMC.** TEA-Seq provides simultaneous measurement of RNA, ADT, and ATAC in the same single nuclei. The RNA- ATAC combined h5 files and ADT count CSV files were downloaded from NCBI GEO (GSM4949911). Customized scripts (available on Github) were used to find the common nuclei in these two files and three different AnnData corresponding to three modalities were generated. We conducted QC for both RNA and ATAC, based on the same cutoffs mentioned above, nuclei with *min_gene* < 300, *min_counts* < 500, *pct_counts_mt* > 20%, *min_atac_peaks* < 1000 will be removed. This step had 8,213 nuclei retained for further analysis. For the RNA, we normalized the RNA UMI count to logarithmic CPTT, we then performed hypervariable gene selection (*top = 3000*), neighbor graph construction (*n_neighbor = 15*), dimension reduction (*n_pc = 15*), and Leiden clustering (r = 1,2,3). For the ATAC, we first normalized the ATAC peak count to logarithmic CPTT, we then performed hypervariable peak selection (top=60000), neighbor graph construction (*n_neighbor = 15*), dimension reduction (*n_pc = 100*), and Leiden clustering (r = 1,2,3). The arguments for n_top_features and n_pc base on both the episcanpy tutorial and dedicated experiments. For the ADT, we conducted CLR normalization using the *scTriangulate.preprocessing.Normalization.CLR_normalization* function, followed by neighbor graph construction (*n_neighbor = 15*), dimension reduction (*n_pc = 15*) and Leiden clustering (r = 1,2,3). Finally, we combine three processed AnnData objects using scTriangulate.preprocessing.concat_rna_and_other function. The combined AnnData serves as the input for the downstream scTriangulate analysis (*win_fraction_cutoff = 0.25, add_metrics = None*). The Azimuth L2 annotation was obtained using post-QC RNA raw UMI count data through the web app (https://app.azimuth.hubmapconsortium.org/app/human-pbmc). All the preprocessing and QC steps were conducted using scTriangulate version 0.9.0 and scanpy version 1.7.2. The Seurat 3-way WNN analysis (RNA, ADT, ATAC) was performed following the official vignette (https://satijalab.org/seurat/articles/weighted_nearest_neighbor_analysis.html). Specifically, we chose 50 Principal Components (PCs) for RNA, 18 PCs for ADT and 2-50 dimensions of latent spaces (Latent semantic analysis) for ATAC, respectively. We adjusted the resolution arguments in the FindClusters function in Seurat Version 4.0 to obtain different clustering results. The totalVI analysis was performed on only RNA and ADT modalities following the official tutorial (https://docs.scvi-tools.org/en/stable/tutorials/notebooks/totalVI.html) using scvi_tools version 0.14.3. Specifically, we chose the top 4000 variable genes as the RNA features and 400 training epochs (default). We performed the Leiden clustering on the totalVI learned joint embedding space and adjusted the resolution arguments in *scanpy.tl.leiden* function to obtain results for a range of clustering resolutions. For evaluation of scTriangulate relative to popular ensemble-clustering algorithms, all 9 independent Leiden clustering results derived from the three independent TEA-Seq modalities (RNA, ATAC, ADT) were provided to the HyperGraph Partitioning Algorithm (HGPA), Meta-Clustering Algorithm (MCLA) and Hybrid Bipartite Graph Formulation (HBGF), using the Python package ClusterEnsembles with default options (https://github.com/tsano430/ClusterEnsembles). Cluster-based similarity Partitioning Algorithm (CSPA) was excluded through the evaluation due to its poor suitability for large datasets.

**Lung cell atlas.** scRNA-Seq and snRNA-Seq cell-population assignment and counts data were obtained from four independent studies (GSE161383, GSE171524, GSE136831, EGAS00001004344). Among these, a lung snRNA-Seq from 9 donors of varying age (newborn, ~3 years old, and ~30 years old)[24] was selected as the base dataset, whereas the remaining were classified into this query dataset using the software cellHarmony. In brief, counts for each dataset were scaled to CPTT and log2 adjusted. For each dataset, reference centroids were initially computed from the author-provided cell-type labels by identifying the top 60 marker genes (MarkerFinder algorithm) for non-diseased control samples (excluding cell-cycle gene enriched clusters). cellHarmony was run in two steps: 1) centroid-based classification of each reference to Wang et al. 2020[24] with a Pearson correlation rho > 0.2, and 2) self-reclassification of Wang et al.[24] to the re-derived markers and centroids from the first classification (MarkerFinder and cellHarmony), to improve the accuracy of the classifications between scRNA-Seq and snRNA-Seq. The same two-step mapping strategy was applied to HLCA (https://app.azimuth.hubmapconsortium.org/app/human-lung-v2), LungMAP CellRef 1.0 (https://app.lungmap.net/app/shinycell-human-lung-cellref) primary Azimuth mappings. Similarly, to assess the quality of scTriangulate myeloid cell population annotations, we compared these specifically to a curated set of Mononuclear phagocytes (MNP-Verse)[29]. Here, we computed gene expression centroids from only healthy lung control samples provided by Mulder et al. (controls from Kim et al.[41]) and mapped the Wang et al[24]. snRNA-Seq dataset using this two-step approach. The obtained annotations and UMAP coordinates[24] were added to the existing AnnData by the *scTriangulate.preprocessing.add_annotation* and *scTriangulate.preprocessing.add_umap* functions, respectively. The RNA raw UMI count data was normalized to logarithmic CPTT. The processed AnnData was used as the input for running scTriangulate (*win_fraction_cutoff = 0.4, add_metrics = 'tfidf5'*). The inclusion of the secondary TF-IDF score was based on a preliminary evaluation with both the default and additional tfidf5 parameters, which better reflected known cell population marker diversity.

**Bone Marrow Cell Atlas.** Alternative scRNA-Seq clustering algorithm results and curated annotations were obtained from our prior benchmarking study[8], using results deposited in the Synapse database (syn26320732). The obtained annotations and umap coordinates were added to the existing AnnData by scTriangulate.preprocessing.add_annotation and *scTriangulate.preprocessing.add_umap* functions, respectively. The RNA raw UMI count was normalized to logarithmic CPTT. The processed AnnData was used as the input for running scTriangulate (*win_fraction_cutoff = 0.3, add_metrics = 'tfidf5'*). Agreement of scTriangulate using results from ICGS2, Monocle3, and Seurat3 with Hay et al. clusters was performed using default parameters (Supplementary Fig. 4d).

### Reporting summary
Further information on research design is available in the Nature Portfolio Reporting Summary linked to this article.

## Data availability
The peripheral blood CITE-Seq CellRanger processed count matrix (h5), scripts, relevant outputs, and metadata have been deposited in Synapse under the accession code "syn26320566". The scRNA-Seq of pediatric AML CD34 + progenitors CellRanger processed count matrix (h5), scripts, relevant outputs, and metadata have been deposited in Synapse under the accession code "syn47980679". Bone Marrow atlas count matrix (mtx) has been deposited in Synapse under the accession code "syn26320732". The scRNA-Seq pre-processed count matrices and genotyping of CD34 + MPN progenitor were downloaded from GEO under the accession code "GSE117825". The scRNA-Seq pre-processed sparse-matrix counts (h5) for PBMC were obtained from the

10× Genomics website (pbmc_10k_v3) and have been deposited to Synapse under the accession code "syn26320659". The PBMC Multiome pre-processed sparse matrix counts (h5) were downloaded from the 10× Genomic website (pbmc-from-a-healthy835donor-granulocytes-removed-through-cell-sorting-10-k-1-standard-2-0-0) and has been deposited to Synapse under the accession code "syn26320419". The PBMC TEA-Seq data (RNA-ATAC combined h5 file) and ADT count (csv file) were downloaded from GEO under the accession code "GSM4949911". Lung Cell Atlas scRNA-Seq and snRNA-Seq count data and original cluster annotations were obtained from four independent studies under the accession codes "GSE161383", "GSE171524", "GSE136831 and "EGAS00001004344". The MNP-Verse reference (.RDS file) was downloaded from "GitHub [https://gustaveroussy.github.io/FG-Lab/]". All other relevant data supporting the key findings of this study are available within the article and its Supplementary Information files or from the corresponding author upon reasonable request. Source data are provided in this paper.

## Code availability

scTriangulate is available as a Python3 package (https://pypi.org/project/sctriangulate). The source code and docker container are available at (https://github.com/frankligy/scTriangulate)[42]. The scripts and data for reproducing the results are available at https://github.com/frankligy/scTriangulate/tree/main/reproduce.

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

## Acknowledgements

We would like to thank Kang Jin, Balaji Iyer, and Daniel Schnell for their valuable feedback on the approach and manuscript. We also thank Kelly Rangel in the CCHMC Gene Expression Core for generating CITE-Seq libraries and Andre Olsson for his assistance. This work was partially supported by the National Heart Lung and Blood Institute (R21HL150678 to H.L.G and N.S., U24HL148865), National Institute of Diabetes and Digestive Health (NIDDK, RC2DK122376 to H.L.G. and N.S.), the Chan Zuckerberg Pediatric Network for the Human Cell Atlas (N.S.) and the Cincinnati Children's Pediatric Cell Atlas Center (N.S. and H.L.G). The Cell Processing Core (RRID:SCR_022623) provided healthy donor cells used in these studies, with related work made possible by support from the NIDDK (U54DK126108).

## Author contributions

G.L. implemented the method and performed evaluations. B.S. generated the CITE-Seq data. N.S., G.L., H.L.G., H.S., and V.B.S.P. designed the study and wrote the paper.

## Competing interests

The authors declare no competing interests.
