## [Peer Review File · Nature Communications]

Decision level integration of unimodal and multimodal single cell data with scTriangulateREVIEWER COMMENTS

Reviewer #1 (Remarks to the Author):

In this article, authors describe a new pipeline called scTriangulate which is claimed as allowing to define the best unsupervised clustering method through combining clustering informations obtained using different datasets and different clustering algorithms.

Major comments:

In Fig. 4, while Adams and Travaglini annotations annotated subsets of cDC, cDC1 and cDC2, scTriangulate split DCs into DC-Lang (are those cDC2/3?), cDC1 and mature DC. Adams defined cDC2, that are defined as IGSF1+ DC by Travaglini, which through scTriangulate, are annotated as Interstitial Mac. Can the authors compare these annotations?

An approach also to evaluate this discrepancy could be to Azimuth these cells on the human MNP-VERSE (Mulder, Immunity 2021) to determine if these cells are cDC2, mono-derived DC or interstitial Macs.

Also, in Fig. 4, while Adams defined cMo and ncMo subsets, scTriangulate regroups them into a single "Mono" population while monocyte experts mostly agree that these 2 human mono subsets should be sub-clustered.

When the authors are talking about new populations, does this mean that a previously unidentified population that is not defined in any of the dataset's analysis can be defined thanks to scTriangulate?

When integrating various datasets, is it possible to also have multiple techniques combined (e.g. RNA+ADT dataset integrated with RNA+ATAC dataset)?

Minor comments:

In the introduction, authors wrote: "1) arbitrary user-defined cluster resolutions". This can be addressed using algorithms such as clustree (<https://github.com/lazappi/clustree>), which allows to define optimal cluster resolution.

Supp Figures are called in random order in the results section. Supp figures should be presented in order in the results.

Supplementary figure 15 has a typo in the heatmap

Supplementary figure 16 b has a formatting issue in the names

Reviewer #2 (Remarks to the Author):

This manuscript introduces a method, scTriangulate, for consolidating different clusterings (i.e., annotations of individual cells) of a single cell sequencing dataset into a single annotation for each cell. The algorithm is agnostic about the source of different cell annotations and the manuscript provides examples of these multiple annotations being derived from separately clustering data from different modalities in multi-omic profiling, or different ways of clustering the same scRNA-seq data. In particular, the manuscript uses scTriangulate to define new subtypes from multimodal data, to perform automated atlas construction, to resolve clonal architectures using genotyping data and to discover splicing-defined subclusters implicated in disease.

The approach (and algorithm) appears to be novel and broadly useful but the method is presented poorly.

In terms of impact, current practice for scRNA-seq analysis doesn't include a quantitative method for

arriving at a best set of clusters, instead often relies on testing out different resolutions and choosing one based on examination of a UMAP or tSNE visualization. If a quantitative method of picking a correct clustering, such as scTriangulate, were to become widely adopted, it would be quite significant. That being said, only minimal quantitative performance benchmarking is in the main text, and it would be useful to clarify how scTriangulate performs compared to other methods for merging multiple clusterings or annotations.

As such, with a well-presented manuscript containing additional validation, this tool could be widely adopted, especially if, as the manuscript claims, there is seamless integration of their framework in the scanpy pipeline.

The novelty of the manuscript is the approach to combine multiple annotations to identify a single consistent set. There are other methods to do this, this is most elegant that we have seen. Again, this paper could better demonstrate why it is a good new addition to these methods by comparing their performance in the main text.

In terms of rigor, the manuscript would benefit from more justification of and a sensitivity analysis on the various choices made for cutoffs and pruning.

Our major concern is in presentation. Though some of us felt that the overall readability and presentation of the paper was satisfactory, we all agreed that the presentation and justification of the algorithm was extremely poor. We found the justification of the algorithm in terms of “cooperative game theory” to be confusing and misleading. In a manuscript where the main contribution is a new algorithm, the main text should contain a clear explanation of the algorithm in plain language. This explanation was missing, and details of how scTriangulate works was difficult to glean from reading the figure captions, methods section, and supplements. Ultimately, we had to consult the source code and rederive the algorithm ourselves in order to understand it. Some of this would be improved by a reorganization of the supplement, though we also found notation errors in the supplement.

However, our effort to rederive the algorithm led to some insights which might be useful in presenting and optimizing the algorithm.

First, it seems like the Shapley value of an annotation (for a cell) for the four different quality scores (Reassign, TF-IDF(5) & (10), SCCAF) is a sum of the Shapley values for each score individually. It also seems like the Shapley of an annotation for a score depends only on its rank among the annotations for that cell. So, we suspect that the algorithm would be a lot simpler and faster if it pre-computed the Shapley value of each of the ranks $1, \dots, |A|$, (where $|A|$ is the number of annotations), put these values in a lookup table, and then quickly calculated the Shapley value of an annotation for a cell by summing the Shapley values associated with each of its ranks across the four quality scores. All this raises the question, that if the Shapley value is just a function of a rank, then why is it the best function, and not just the rank itself? Also, I wonder if each annotation could be given an overall quality score (like an Elo rating from chess) based on the per-cell comparisons, and then the annotations for each cell could be assigned based on this Elo rating.

Below we have listed some clarifications (of the issues raised above) and additional comments and concerns:

- Fig. 2: Small ADT-2 cluster in Fig. 2a isn't marked with ADT contribution in 2e, but instead has a high RNA contribution.
- Some subfigures of Fig. 2 not referenced in the main text, lacking explanation of their relevance apart from the legend itself.
- No study of robustness of integration results to parameter choice (for e.g., default thresholds mentioned in line 112)
- Label pruning and assignment to the nearest neighbor seems ad hoc and not a general consistent solution. In particular, it seems to break the philosophy of using the quality scores to assign annotations. Why not just default to the next highest scoring annotation for each cell?

- Error in Fig. 4 legend - ...corresponds to source dataset annotations in panel b*
- Error in Fig 5. Legend d) Expression heatmap of final scTriangulate*
- Abbreviations such as GoT should be expanded (e.g., for those not familiar with clonal genotyping literature)
- Exceedingly brief discussion of relevance in the field and future directions
- It would also be interesting to seeing how the method performs when provided with clusterings from a single modality at different resolutions (i.e., numbers of clusters). Will it always tend towards a higher or lower resolution or a combination thereof?
- It's hard to understand how the scTriangulate actually works. The methods section is very unclear; variables seem to be describing multiple things, and understanding what Algorithm 1 (which computes the difference in team score within the overall Shapley Value calculation) was doing took several readings and examination of the python implementation on github.
- The value provided by using the Shapley Value is not obvious. The 'games' being played for each cell seem to mostly be finding the highest ranking annotation across multiple metrics in a very complicated way – is this really all that different than just summing the ranks across all the quality scores and choosing the best sum? In particular, the actual collaborative element here is not clear to me.
- The comparisons to alternate methods also need to be expanded. In figure 2f, where you compare scTriangulate's agreement with a set of reference labels to that of other methods, the figure caption states that you are comparing with both Seurat WNN and single-modality Leiden clusters, but in the figure, there is only scTriangulate and different Seurat WNN resolutions. If by 'single-modality Leiden clusters' you mean those that were used in the scTriangulate pipeline, this is not made clear. Also, no justification is given for the choice of Seurat WNN clusterings as your only point of comparison (outside of the supplemental figures). I think the comparison against existing ensemble methods is important enough to be in the main paper instead of the supplement. Also, the metrics used for these comparisons (homogeneity, completeness, & v-measure) are neither explained or justified in this paper.
- Confidence is misspelled in line 98
- The comma on line 235 is not necessary, and the sentence is overall quite clunky. Maybe it could be split into two sentences?
- In equation (8) in the supplement, what is indexed by "i"? Should this be "a"? This same error is in the pseudo-code. This error is particularly bad because "i" is used to index cells immediately after Algorithm 1
- What is the role of the 0.01 in the pseudo-code. Is this just a magic number to resolve tied ranks? If so, this seems arbitrary and not robust.
- I couldn't understand how the "Reassign" score was computed based on its description. Please clarify this.

Reviewer Response Summary

We thank the reviewers for their careful consideration of our manuscript and noted appreciation of the novelty of this approach. Here, we provide a major revision of the original manuscript to thoroughly address the reviewer concerns and questions.

Specifically, we provide:

1. Broad restructuring of the manuscript figures and associated text.
2. In depth mathematical and conceptual justification of the approach using plain language.
3. Extensive new benchmarking studies (real and simulation).
4. Comprehensive sensitivity analyses for all tunable parameters in the software.
5. New tunable parameters, quality metrics, software environments and expanded documentation.

Importantly, our significantly expanded real and simulation benchmarking, demonstrate the superior and unique functionality of our approach for single-cell dataset integration from diverse analytical and multimodal sources. Hence, we are confident in our original claim that game theory provides a novel and vastly improved solution to flexibly integrate single-cell data from nearly unlimited sources. To echo comments from Reviewer 2, we believe this approach has the strong potential to be widely adopted in the realm of single cell genomics due to its unique ability to resolve conflicts and distinguish real signal from noise.

Reviewer #1 (Remarks to the Author):

Major comments:

1. In Fig. 4, while Adams and Travaglini annotations annotated subsets of cDC, cDC1 and cDC2, scTriangulate split DCs into DC-Lang (are those cDC2/3?), cDC1 and mature DC. Adams defined cDC2, that are defined as IGSF1+ DC by Travaglini, which through scTriangulate, are annotated as Interstitial Mac. Can the authors compare these annotations? An approach also to evaluate this discrepancy could be to Azimuth these cells on the human MNP-VERSE (Mulder, Immunity 2021) to determine of these cells are cDC2, mono-derived DC or interstitial Macs.

We thank the reviewer for positive comments and careful evaluation of its results. To address these noted concerns, we have: 1) clarified the statistical differences that underlie scTriangulate decision making in lung for noted and other populations, 2) compare scTriangulate results to new lung/immune atlases and 3) assess these assignments based on marker genes.

First, an important note is that there is much debate about which proposed resident immune and non-immune populations in the lungs exist. Indeed, this debate has necessitated the construction of new lung cell atlases by the Human Lung Cell Atlas (HLCA) initiative, LungMAP, in addition to those included in this research study (Travaglini, Melms, Adams, Wang). In running scTriangulate, we assume each of these prior atlases are inherently valid, but that specific studies will be biased in the separation and annotation of specific lineages (author bias).

A cross-consortium working group was recently established to address these concerns, initiated by the LungMAP, to resolve these conflicts through expert curation (Cell Cards initiative - <https://www.lungmap.net/cell-cards/>). This includes the creation of an integrative single-cell atlas called Lung CellRef (<https://www.biorxiv.org/content/10.1101/2022.05.18.491687v1>).

scTriangulate is designed to identify the most stable combinations of clusters from diverse independent sources. Here, multiple stability metrics are compared for each cell population and each individual cell to find the source cluster/annotation that has the best statistical measures of stability. As noted by the reviewer, Wang et al. DCs and a subset of Mono's, were split into Travaglini pDC and Adam's mature DC, cDC1 and DC-Lang, all evidenced by unique marker genes (**Supplemental Figure 8**). In the case of Wang Interstitial Macrophages, the reviewer notes that this population is annotated as four different names by different research groups, in each of the four compared atlases (Mono-derived Macs, interstitial Macs, IGSF21+ DC and cDC2). Such divergent names make it difficult to know which name is most accurate and whether scTriangulate happened to select a biologically more appropriate name. As scTriangulate only assesses statistical stability as opposed to considering the actual name, we will first look at the statistical considerations. Comparison of the stability metrics for these choices show there are clear benefits for both Wang and Travaglini predictions, which are assigned equivalent ranks for re-assign and TF-IDF-10 (**Reviewer Figure 1a-c** and **Supplementary Figure 8d**). Here, the scores slightly vary, however, the program ranks these equally since they are "very close" (<0.01 difference in scores - see **Reviewer 2 response 5**). The major scoring differences are in SCCAF and TF-IDF5 for Wang and Travaglini, with Wang selected as the final winner in the game (determined by the Shapley value). TF-IDF5 translates to the relative specificity of the 5th top unique marker in the different competing clusters (e.g., Adams, Wang, Travaglini). Since Shapley more strongly weights TF-IDF5 in the game, the Wang Interstitial Macrophage is selected as the winner, without TF-IDF5, Travaglini would be the clear winner. Both options are valid in the software, however, inclusion of TF-IDF5 will favor smaller clusters with greater marker gene specificity.

To address which annotation is actually correct, we can compare these results to multiple independent reference atlases, as recommended. Here, we considered the: 1) LungMAP CellRef 1.0 atlas with 48 populations (Azimuth), 2) Human Lung Cell Atlas (HLCA) with 58 populations and 3) MNP-Verse with 19 immune populations (finest resolutions). In general, our mapping strategy (initial alignment with secondary refinement - **Methods**) yielded generally clear links between scTriangulate results and the corresponding mappings onto Wang et al, based on UMAP visualization (**Reviewer Figure 1d-f**). Importantly, the more recent HCLA and CellRef atlases confirm many of the scTriangulate proposed subdivisions in diverse lineages (e.g., epithelial, endothelial, myeloid, B-cells) and potentially resolve conflicts in these two atlases (e.g., identification of platelets). As shown, Wang Interstitial Macs, were predicted by HLCA and CellRef to be Interstitial Macs, with infrequent contamination of Classical monocytes (HLCA) or moDCs (CellRef). Here, scTriangulate had improved agreement with both HLCA and CellRef versus ensemble clustering approaches (**Figure 5g**). Using MNP-Verse as a reference resulted in more heterogeneous cluster assignments, with Wang Interstitial Macs annotated as: Proliferating cells, Mac-13, DC2-DC3-15, Mac-23, Mac-8, among others (**Reviewer Figure 1f,g**). Hence, there is evidence based on this analysis that the large majority of cells in Wang Interstitial Macs, are indeed Interstitial Macs, with some potential heterogeneity

of this cluster intermixing with DCs, that was not resolved with the input reference annotations to scTriangulate. This is further evidenced by expression of the DC marker *CLEC10A* in a subset of Wang Interstitial Macs, which partially overlaps with that of the Macrophage marker *MERTK* (**Reviewer Figure 1h**). As also noted by the reviewer, DC-Lang's here would also be suggested to be renamed to cDC2's based on these new reference mappings in the final scTriangulate atlas. We now present these additional mappings as evidence of these clusters and discuss the challenge for proper label annotation in the results and discussion.

Results: “Comparison of scTriangulate predictions to a cross-tissue mononuclear phagocytes atlas called MNP-VERSE, supports our finding of expanded DC and macrophage heterogeneity in the lung (**Supplementary Figure 8b**)³⁰. While in many cases, more granular cluster predictions are rejected by scTriangulate (e.g., ncMono and cMono in Adams) (**Supplementary Figure 8c**), we cannot rule out the possibility that such populations will be better resolved in larger patient cohorts or considering additional modalities. Further, as the naming of populations will often vary between studies (example in **Supplementary Figure 8d**), expert curation of the scTriangulate integrated labels will always be required. To further assess the quality of scTriangulate cluster predictions and names, we compared these to two independent integrated lung atlas annotation efforts (Human Lung Cell Atlas (HLCA) and CellRef, **Methods**) (**Figure 5f** and **Supplementary Figure 8e**). In addition to identifying confirmatory support for many of scTriangulate's subdivisions (epithelial, endothelial, myeloid, B-cells) and names, scTriangulate had improved agreement with independent lung atlases over those obtained with ensemble clustering (**Figure 5g** and **Supplementary Figure 8f**).”

Discussion: “Further, our results show that the integration of prior obtained annotations from lung or bone marrow identifies improved cellular identities that can only be obtained from the combination of multiple independent sources. As such, scTriangulate has the strong potential to automate cell atlas creation through the integration of different proposed annotations and new clustering results. These can include new populations which arise from partially overlapping annotations from different sources. An important remaining challenge of such analyses is the naming of populations, which remains largely non-standardized (see **Supplementary Figure 8**).”

Reviewer Figure 1. Resolving heterogeneity and cell-type identity in lung with scTriangulate. a) Prior-published cell-annotations and UMAP from Wang et. al. b) scTriangulate integrated final clusters in lung, considering Wang, Melm, Adams and Travaglini references (cellHarmony). c) scTriangulate considered stability metrics for Wang Interstitial Macs, compared specifically to overlapping Adams and Travaglini aligned cluster definitions. d-f) Cell atlas annotation projected onto Wang et al. snRNA-Seq from (d) HLCA, (e) CellRef and (f) MNP-Verse (myeloid-specific reference). g) Composition of Wang Interstitial Macs relative to MNP-Verse projected label assignments. h) Abundance of DC (*CLEC10A*) and Macrophage (*MERTK*) marker genes in Wang et al. i) Comparison of scTriangulate stability metrics for Wang Monocytes, relative to Adams cMono and nMono. Adams' populations have improved TF-IDF5 scores (more specific markers) but are less stable by other evaluated metrics. Consideration of only TF-IDF scores would hence likely result in selection of cMono and nMono as the most stable populations.

2. Also, in Fig. 4, while Adams defined cMo and ncMo subsets, scTriangulate regroups them into a single “Mono” population while monocyte experts mostly agree that these 2 human mono subsets should be sub-clustered.

This is helpful to point out as we do not wish to misstate that scTriangulate populations in this lung study are comprehensive, as they are not, for various and important reasons unrelated to the algorithm. As noted in **Reviewer 1 response 1**, the most critical determinants of cell-type validity are prior functional and in depth single-cell studies probing specific cellular compartments of the lung (i.e., CellCards). In our demonstration analysis for creating cell atlases, we use a single snRNA-Seq study as our base for then assigning cell-type annotations based on independent atlases for triangulation. There are several caveats to this approach:

1) Technical considerations - Wang et al., profiles nuclei from a dozen donors spanning multiple developmental ages. Nuclei profiles will differ from conventional cytoplasmic RNA profiles (potentially impacting label transfer) and that some populations will be absent or under-represented due to anatomical and cell depth limitations, which are inherent in most all lung single-cell studies. Cell surface markers (e.g., CITE-Seq) are not present in these studies further limiting the ability to accurately immunophenotype cells.

2) Atlas considerations - While Wang et al. was selected due to its reasonable depth of cells (>40,000), donors (n=12) its decent representation of distinct lineages, and developmental timepoints (n=3), it only has “healthy” cells, does not consider additional single-cell chemistries or platforms and does not integrate hundreds of samples in diverse airway locations. Hence, it will not be sufficiently comprehensive to detect certain rare populations (e.g., ionocyte, neuroendocrine, distinct T-cell subsets). Other studies intentionally bias towards the selection of specific lineages to resolve these challenges and extensively integrate across studies while attempting to address batch effects. While such integrated atlases could be provided to scTriangulate, this will depend on proper batch and technology effect considerations/normalization, which are outside the scope of the current study.

3) Reference annotation limitations - Our analysis only included four source reference annotations, where cMo and ncMo were not the most stable options (**Reviewer Figure 1i**). It may be that improved source annotations or sub-clustering will better resolve, which is important for scTriangulate users to consider (supplying a range of unsupervised clustering resolutions in addition to label transfer).

We now attempt to clarify these challenges and study-level limitations, which remain important considerations for the creation of comprehensive reference cell atlases (see manuscript text in **Reviewer 1 response 1**).

3. When the authors are talking about new populations, does this mean that a previously unidentified population that is not defined in any of the dataset’s analysis can be defined thanks to scTriangulate?

The reviewer notes two important points for clarification. First, all “new” populations mentioned in manuscript refer to clusters derived from different clustering resolutions that meet the following conditions: 1) these PBMC cell-populations do not exist in the annotated “silver-standard” reference (PBMC Azimuth, derived from CITE-Seq), 2) the same “new” populations can be reproducibly identified in distinct scRNA-Seq, CITE-Seq, TEA-Seq and/or multiome datasets (based on similar marker of expression), 3) they have prior evidence in the literature, to indicate they are not common technical artifacts. Hence, such populations do not represent novel cell types, but rather cell populations not present in Azimuth. Specific examples referred to in this paper are: 1) classical, intermediate, and inflammatory CD14 monocytes, 2) *CASC15*-defined innate lymphoid cells (ILC) and 3) effector memory, naïve and central memory CD4 T-cell subsets (**Figure 4d,e** and **Supplementary Figures 5e,f** and **6h**). In each case, existing literature evidence their existence, and multi-resolution mix-and-match analysis further demonstrates they can be found from one or more modalities and are “stable”.

Second, we want to emphasize that scTriangulate indeed is capable of identifying biologically meaningful “new” subpopulations.. This occurs when only a subset of cells within an input broad cluster are found to be stable, resulting in a smaller, coherent group of cells. In many cases, such clusters will be automatically removed, as by default, the program removes any stable clusters with only a small proportion of the original cluster cells retained (<25%) (**Methods**). In cases where such “new” clusters are retained, scTriangulate suggests these represent biologically valid sub-clusters. Clear examples of this can be seen in the lung integration analysis (**Reviewer Figure 1b**). For example, Wang, Adams, Melms and Travaglini annotate a single cluster named Ciliated cells, which is not subdivided into distinct sub-populations. In the final scTriangulate result, there are two ciliated cell populations, one from Adams and one from Melms. This is because some cells were more stable for Melms and others more stable for Adams. In this case, there is evidence the split is real and appropriate, as HLCA and CellRef annotate a new population of ciliated cells (Deuterosomal) missing in our prior four references. A second example are subsets of MAIT cells defined by the integration of different resolutions and different CITE-Seq modalities in PBMC (see below). We now clarify both of these points in the results and discussion (examples in **Reviewer 1 response 1**).

Results: “Beyond these reference Azimuth populations, scTriangulate uncovered distinct subtypes of monocytes and MAIT cells, defined by the unique combination of RNA and ADT features (Supplementary Figures 4h-j). Importantly, these distinct subsets of MAIT cells could only be derived through the integration of ADT and RNA clusters, as no source resolutions identified these populations.”

4. When integrating various datasets, is it possible to also have multiple techniques combined (e.g. RNA+ADT dataset integrated with RNA+ATAC dataset)?

The problem mentioned by the reviewer, is both an exciting natural extension of the scTriangulate approach and a substantial analytical challenge for the field. In short, the answer is yes, if an upstream program can effectively perform the cross-dataset integration and account for batch effects among features. To be more specific, there are currently three types of unmatched multimodal integration analyses we are aware of: 1) integration of cells to a common

latent space from different datasets using only the matching modality, 2) integration of cells to a common latent space from different datasets for multiple overlapping modalities, or 3) imputation of missing modalities from one assay. Typically, if a researcher wants to integrate CITE-Seq with another multimodal assay (e.g., TEA-Seq, multiome), they would only integrate based on RNA (senario #1). This is commonly done with tools such as Harmony, MNN, Seurat CCA, but could also potentially be accomplished with new approaches such as BRIDGE-integration in Seurat, which uses dictionary learning. For multimodal integration of different multimodal techniques (senario #2), it is necessary to integrate based on any overlapping modalities. In the case of TEA-Seq (ADT+RNA+ATAC), the assay supports overlapping ATAC and RNA features with multiome or ADT and RNA features with CITE-Seq. There are multiple new graph neural network and deep-learning based integration approaches that have been recently proposed to solve this problem, but they have not been extensively benchmarked, with only limited preliminary code available (e.g., Multigrade). Another approach, is to integrate different single-cell datasets based on a single-modality (e.g., RNA) and then impute the missing modalities in either dataset (senario #3). This is the conceptual goal of scARCHES, but has only been demonstrated with scRNA-Seq and CITE-Seq integration/imputation, producing results based on the integration of a single modality. Finally, any integration, would also need to effectively correct for any technology-specific batch effects, which are significant between nuclear assays (TEA-Seq, multiome, snRNA-Seq) and cytoplasmic assays (scRNA-Seq, CITE-Seq, MAS-Iso-Seq, splicing). Overcoming such objects remains an extremely challenging and important analytical challenge in the field of single-cell genomics which is outside the scope of the current manuscript, but is an ongoing objective of scTriangulate development, as now noted in the discussion. Nonetheless, users able to resolve this integration with new or emerging tools could readily apply scTriangulate on the resultant label transferred cluster assignment, post-integration and batch effects removed imputed feature data.

Discussion: “In addition to the analysis of new technologies, it will be important for this and similar approaches to support the integration of multiple samples of the same modality and even across different single-cell multimodal or spatial technologies. We aim to support such features in the future, through the incorporation of innovative new computational strategies.”

Minor comments:

5. In the introduction, authors wrote: “1) arbitrary user-defined cluster resolutions”. This can be addressed using algorithms such as clustree (<https://github.com/lazappi/clustree>), which allows to define optimal cluster resolution.

We now clarify this point and provide additional simulation data to support the central hypothesis of this paper. The hypothesis we propose is that not only are individual resolutions arbitrary, but that the most discrete and stable populations require consideration of multiple resolutions through their integration and testing. Similarly, when considering multimodal datasets, each modality should have separate clusters derived/projected at multiple resolutions. Hence, we propose that no single resolution for a single-modality is sufficient to obtain optimal solution for the entire dataset. To test this hypothesis, we previously benchmarked multiple

clustering algorithms, both single-modality (scanpy) and multimodality clustering (WNN, TotalVI) for a diverse range of resolutions and show scTriangulate (mixing and matching cluster predictions from different source resolutions/modalities) provides the most accurate predictions, relative to a Azimuth PBMC reference (**Figure 3g**). We further benchmarked such methods against ensemble clustering approaches, which only consider the similarity of the input clustering solutions, without considering stability or even underlying features in the data. Our newly included simulation data explicitly tests our hypothesis, using a verifiable ground-state truth (known transcriptionally distinct cell populations) simulated from Splatter (**Reviewer Figure 5 and Figure 2**). Here, we hold out the simulation dataset ground-state truth cluster solution and test different solutions, each with under-clustering or over-clustering of different populations. While ensemble clustering fails to identify the correct solution, scTriangulate perfectly matches the simulation. Hence, we show that when different stable populations are present in different resolutions, scTriangulate can uniquely determine an optimal integrated solution.

One additional point of clarification is that approaches such as clustree do not actually identify which resolutions are “optimal”, but only visualize splits between clusters in the data as illustrated below (**Reviewer Figure 2**). Hence, running clustree on multiple leiden resolutions produces no clear tunable parameters to optimize an analysis.

Reviewer Figure 2: Ambiguous determination of optimal resolutions for subcluster analysis using standard approaches. Output of clustree on PBMC scRNA-Seq with seven independent Leiden (scanpy) resolutions.

6. Supp Figures are called in random order in the results section. Supp figures should be presented in order in the results.

Thank you for correcting. Figure callouts have been updated, consolidated and reordered to ensure sequential callouts.

7. Supplementary figure 15 has a typo in the heatmap.

This was a PDF conversion issue that has now been corrected.

8. Supplementary figure 16 b has a formatting issue in the names.

This was a PDF conversion issue that has now been corrected.

Reviewer #2 (Remarks to the Author):

This manuscript introduces a method, scTriangulate, for consolidating different clusterings (i.e., annotations of individual cells) of a single cell sequencing dataset into a single annotation for each cell. The algorithm is agnostic about the source of different cell annotations and the manuscript provides examples of these multiple annotations being derived from separately clustering data from different modalities in multi-omic profiling, or different ways of clustering the same scRNA-seq data. In particular, the manuscript uses scTriangulate to define new subtypes from multimodal data, to perform automated atlas construction, to resolve clonal architectures using genotyping data and to discover splicing-defined subclusters implicated in disease.

1. The approach (and algorithm) appears to be novel and broadly useful but the method is presented poorly.

We thank the reviewer team for carefully reviewing our code, documentation and methods to determine which methods are not described in sufficient depth and identify methods that should be further individually benchmarked against other reasonable alternatives. In this revision, we have reviewed each step of our methods and documentation descriptions, with the aim of providing additional rationale, background context, pseudocode where missing and benchmarks where possible. Specifically, we now clarify: 1) existing multimodal integration approaches (advantages, limitations), 2) our central hypothesis (no single-resolution, algorithm or modality is sufficient to produce an optimal stable solutions for all populations), 2) the need for integration of unsupervised and/or supervised results for known and novel modalities, 3) the rationale for the use stability metrics, 4) the rationale for the use of game theory to solve this integration challenge, 5) provide additional details on each step of the algorithm (joint feature modality consideration, PCA ranking, pruning, reclassification) in the results and 6) extensively describe the novel capabilities of this approach. We have further provided additional benchmarking and highlighted such benchmarking in the primary figures to emphasize reliability of the approach (**Figures 2 and 3**). Finally, we have provided a more detailed discussion, which includes high-level conclusions, statistical consideration, future opportunities and compatibility with new modalities/technologies, such as spatial transcriptomics. Finally, we have used the provided feedback to add new vignettes and code examples to our online documentation.

2. In terms of impact, current practice for scRNA-seq analysis doesn't include a quantitative method for arriving at a best set of clusters, instead often relies on testing out different resolutions and choosing one based on examination of a UMAP or tSNE

visualization. If a quantitative method of picking a correct clustering, such as scTriangulate, were to become widely adopted, it would be quite significant. That being said, only minimal quantitative performance benchmarking is in the main text, and it would be useful to clarify how scTriangulate performs compared to other methods for merging multiple clusterings or annotations.

We thank the reviewers for this important feedback. While considerable benchmarking was performed in the prior iteration, much of this was buried in Supplementary Figures, with insufficient callouts in the main text. We have addressed this concern by: 1) moving our extensive benchmarking for single- and multi-modality evaluations to the main Figures where possible, 2) providing improved callouts and descriptions of the benchmarks, 3) statistical evaluations and rationale for these benchmarking methods and 4) simulation-based testing and benchmarks. For these evaluations, we not only consider ensemble clustering, but relative performance of scTriangulate relative to multiple software resolutions, modalities (RNA, ADT, ATAC, splicing), algorithms (scanpy Leiden, Seurat CCA, ICGS2, Seurat WNN, TotalVI) and integration methods (MLCA, HGPA, HBGF) and hence, consider our benchmarking to be both rigorous and extensive. For cluster evaluation, we now apply homogeneity, completeness and V-measure in addition to two other commonly used cluster metrics, Adjusted Rand Index (ARI) and Adjusted Mutual Information (AMI). These methods are now more clearly described and justified in the **Methods**. Our newly included simulation data explicitly tests our hypothesis that scTriangulate is needed to resolve the most stable populations across multiple as opposed to a single clustering solution, using a verifiable ground-state truth (transcriptionally distinct cell populations) simulated from Splatter (**Reviewer Figure 5** and **Figure 2**). Here, we hold out the simulation dataset ground-state truth cluster solution and test different solutions, each with under-clustering or over-clustering of different populations. While ensemble clustering fails to identify the correct solution, scTriangulate matches the simulation. Hence, we show that when different stable populations are present in different resolutions, scTriangulate can uniquely determine an optimal integrated solution. Here, scTriangulate outperforms other algorithms, as we would hypothesize given the varying performance of different resolutions to capture unique cellular subsets and the inability of ensemble approaches to assess stability.

3. As such, with a well-presented manuscript containing additional validation, this tool could be widely adopted, especially if, as the manuscript claims, there is seamless integration of their framework in the scanpy pipeline.

That is correct, scTriangulate is seamlessly integrated with scanpy (<https://github.com/frankligy/scTriangulate#tutorials-and-installation>). In addition to the extensive tutorial and documentation, we provide a minimum working example in our GitHub/ReadMe to quickly get new users running the code and producing outputs. We have further updated the logging and multiprocessing module to make it more efficient and standardized. We now additionally provide a docker container to enable improved reproducibility in a version controlled manner for end-users and added initial support for spatial transcriptomics analysis (integration of spatial, cellular and molecular features).

Discussion: “Looking forward, we anticipate extension of this approach beyond single-cell genomics to more complex settings, such as spatial transcriptomics, in which the notion of molecular modalities can be extended to spatial position within a tissue, cellular niches, subcellular localization, cellular interactions, cell morphology, and other qualitative and quantitative readouts enabled through image-based analyses. *scTriangulate* is extendable to such approaches and already possesses an interface to the python spatial transcriptomics analysis package *Squidpy*.”

4. The novelty of the manuscript is the approach to combine multiple annotations to identify a single consistent set. There are other methods to do this, this is most elegant that we have seen. Again, this paper could better demonstrate why it is a good new addition to these methods by comparing their performance in the main text.

Please see **Reviewer 2 responses 1 and 2**, which address the concerns related to the novel capabilities of *scTriangulate* over alternative methods and benchmarking. Importantly, unlike ensembl-based approaches, *scTriangulate* evaluates cluster importance for distinct modalities, considering multiple stability metrics, in a cooperative manner (see **Figures 2, 3 and 5**).

5. In terms of rigor, the manuscript would benefit from more justification of and a sensitivity analysis on the various choices made for cutoffs and pruning.

We thank the reviewer for helping us clarify the rigor and reproducibility of the approach. In addition to our prior and new simulation benchmarking, we now perform a systematic sensitivity analysis considering all tunable parameters. For evaluation, we have varied each of these parameters when applied to our five non-synthetic PBMC single-cell and single-cell multimodal dataset. Specifically, we assessed:

- (a) Minimum number of cells per cluster (Cluster Size)
- (b) Thresholds for determining rank from individual stability metrics (Offset)
- (c) Optional rescaling of stability ranks (Rank Strategy)
- (d) Inclusion of additional stability metrics (Stability Metric Addition)
- (e) Alternatives to Shapley (Annotation Importance)
- (f) Cluster pruning percentage threshold (Pruning)

Cluster Size: Assessments of stability in extremely small clusters is inherently problematic, as reclassification scores will be biased to a few cells driving consistent differences. Hence, by default, the program excludes clusters with fewer than 10 cells (`abs_thresh` parameter). Here we assess a range of cutoffs.

Offset: Prior to computing annotation importance, each stability score is converted into a rank, where each annotation is assigned a different or equivalent rank. If a cell corresponds to 5 different possible annotations, the annotation with the best score will receive a rank = 5 (default convention of `scipy.stats.rankdata`), and the others will be ranked in descending order accordingly. When computing, an offset value is considered (default), such that very similar

stability scores will have equal ranks. The use of an offset is inspired by General Linear Models (GLM), where an offset term is used to account for residual variation that is disproportionate and cannot be explained by the regression model itself. An offset was introduced into our codebase early in development, as we frequently observed situations where prior validated cell-populations were NOT selected by scTriangulate, as a result of stability scores that differ by less than 0.01 (scores typically range from a value of 0-2). This fixed value parameter was considered reliable, as all stability metrics produce scores in a fairly similar range (adjustments may be required for new scores outside this range). Without the offset, one annotation (e.g., verified rare population 1) would be out-selected by a second annotation (e.g., broader population 2), although both have almost equivalent scores. Here we assess a range of offset values.

Rank Strategy: After initial ranks are obtained (after offset consideration), the final values provided to the Annotation Importance method are optionally scaled. Specifically, the two existing scaling options are “classic” and “all_or_none” (default). By default, the program re-scales all ranks for each cell and for each stability metric in a winner takes all manner (all_or_none), where only the top rank choice(s) retain its rank (all others set to 0). If multiple annotations have the top rank (considering the offset), then each of these annotations will be assigned the same non-zero rank. The rationale for this option is that the best performers will rise to the top, producing more definitive ranking for downstream steps. This option was suggested as an important factor in our initial methods evaluation, after examining ranks and Shapley values for dataset with known rare experimentally validated cell populations. Conversely, “classic” will retain all of the original ranks, post-offset consideration.

Stability Metric Addition: In addition to the use of SCCAF, reassign and TF-IDF10, the software can consider a fourth stability metric called TF-IDF5 (default). This metric is identical to TF-IDF10, but assesses the relative marker specificity score in the 5th top ranked feature, versus the 10th. In our Bone Marrow evaluations, this additional parameter was needed to identify previously experimentally validated cell populations (e.g., well-defined HSCP subsets), as described in the **Results** and **Methods**. Hence, this option will often prioritize small populations with greater marker gene specificity than alternatives, but that may be less stable (e.g., metastable transitional cell states). Here, we evaluate scTriangulate with and without TF-IDF5.

Annotation Importance: The Shapley value is used in scTriangulate for fair credit allocation of the ranks derived from different stability metrics for each cell and player. Shapley applies coalitional iteration to compute the marginal contribution for each player. As this method tests the benefit of all possible combinations of players (cooperatively), in an iterative manner, compute time is relative to the number of players provided considered. In considering alternative annotation importance methods (e.g., competitive game theory, Nash theorem), there are important computational complexity and solution incompatibilities that prohibit their use in this application (see details in **Reviewer 2 response 7**). An alternative to Shapley, is simply summing the ranks from the different stability metrics (after optional rank scaling). Here, we test simple rank-based scoring to Shapley.

Pruning: As scTriangulate assigns stable populations at the level of individual cells, an optional pruning step was introduced to exclude cell populations with few cells retained relative to the source cell-to-cluster assignments. First, we assume that if a large initial cluster of cells exists, but only a small minority of cells with that annotation are considered to be "stable", that the overall predictions is more likely to be unreliable. For example, if a 1,000 cell cluster is initially annotated as Macrophages, but only 5% of the cells at the end retain this annotation, the source annotation should not be considered reliable or specific to this small fraction of cells. For this reason, we recommend the default pruning threshold of 25%, to exclude such predictions and re-assign the label of these cells to the nearest neighbor, which is likely to represent the dominant annotation replacing the original Macrophage label. There are different hypotheses to why this might happen, assuming that our game-theory approach is perfect. First, it is assumed that scTriangulate retains this small proportion of cells because they have unique molecular features from their neighbors. However, our null hypothesis is that these are more likely to be technical artifacts (e.g., high mitochondrial expression, dying cells). Alternatively, this small subset of cells may represent a valid "activation" state of the Macrophages and could be valid. See **Reviewer 2 response 11**, for additional rationale. Hence, we tested different pruning thresholds in this evaluation.

Evaluation Results:

- a) Cluster Size: Considering 11 different cutoffs (1-50 cells), we observe no consistent appreciable performance gain or loss, for the different evaluation datasets (**Reviewer Figure 3a**). Notably, the default threshold of 10 produced consistent reliable results.
- b) Offset: Considering 5 different tolerance thresholds (ranging from 0-0.1), we again find no appreciable consistent performance gain or loss (**Reviewer Figure 3b**).
- c) Rank Strategy/Annotation Importance: Considering all_or_none or classic rank scaling in conjunction with Shapley, only minimal performance differences were observed, with the exception of TEA-Seq, in which all_or_none resulted in a more substantial performance gain over classic. However, considering an alternative annotation importance method (simple rank-based), produced highly variable results for all_or_none or classic. Specifically, rank-based selection ineffectively identified reference-cell populations (Azimuth) when used with all_or_none, while classic with rank-based produced valid predictions, similar to the software defaults (**Reviewer Figure 3c**).
- d) Pruning: Considering a pruning threshold of 0-60% (60% of cells from the original cluster must be retained to prevent pruning and reclassification), we find that more pruning is predicted to always improve performance. However, minimal-to-no performance improvement was gained beyond a pruning threshold of 40%. Hence, pruning tends to result in improved concordance (V-measure) with established reference clusters, but with minimal performance gain beyond a threshold of 40% (Azimuth) (**Reviewer Figure 3d**).
- e) Stability Metric Addition: Considering four subtly different cell populations simulated with splatter, we find that inclusion of a secondary TF-IDF score (TF-IDF5) identifies all simulated clusters whereas excluding this option rejects the selection of subpopulations (**Reviewer Figure 4** and **Supplementary Figure 2**). Hence, use of additional stability

metrics will prioritize the selection of stable subclusters with unique biologically informative features.

Based on these analyses, we conclude that the default options in scTriangulate are optimal for the real and simulation datasets evaluated. As simulation datasets will be limited to distinguish diverse biologically informed cell-population differences, these results are primarily informed from multimodal and unimodal blood datasets. As the use of game-theory represents the central algorithmic innovation of scTriangulate, it is important to clarify why its use as a default method in scTriangulate is warranted. First, Shapley produces improved or equivalent results to all rank-based analyses, independent of the rank scaling method applied, whereas rank-based analyses are dependent on the ranking strategy. We attribute this observation to the fact that Shapley iteratively tests all combinations of players in a comprehensive manner, as compared to rank-based which is subject to stability metric variance. Nonetheless, given the computational efficiency of rank-based analysis (complexity increase exponentially $2^{**}n$ for Shapley), we have made this method the default (in conjunction with the classic rank strategy), when the number of players is greater than 15. As recommended, we now also include a metric analogous to the Elo rating, that is represented as a normalized Shapley value, to determine which cluster is likely more stable (quality score) (see details in **Reviewer 2 response 7**). Using this quality score, users can effectively rank different players for a given dataset (e.g., cluster resolutions, projected labels, modalities). To allow the user to change the evaluated metrics and other settings after analysis has been run, we have added a function called salvage_run, to store all scTriangulate results and re-run with different options. Our codebase and documentation have been updated accordingly.

Reviewer Figure 3. Sensitivity analysis for distinct tunable parameters in scTriangulate.

The performance of different scTriangulate tunable parameters were tested for the blood single-cell genomics datasets evaluated in this manuscript (unimodal and multimodal) (Methods). Each bar indicates the correspondence of clusters in scTriangulate when compared to the PBMC Azimuth reference (quantified by V-measure). a) Comparison of cutoffs for reliable minimum number of cells in a source cluster (Cluster Size) for all evaluated annotations, ranging

from 1-50 cells. b) Comparison of different stability metric rank tolerance (offset) thresholds, ranging from 0.0-0.1. c) Comparison of different annotation importance strategies (Shapley and simple Rank-based), for different rank prioritization approaches (all_or_none, classical). All_or_none = winner takes all strategy. Classic = unadjusted rank strategy. d) Comparison different cluster pruning thresholds, ranging from 0-60% of cells retained from the original parent cluster.

Reviewer Figure 4. Benefit of additional stability metrics in the game. a) As a ground-state truth to assess the specificity of predictions by scTriangulate, we simulated scRNA-Seq data for two highly distinct simulated populations (c1 and c2) and four subtly different subclusters (c3, c4, c5, c6) (Methods). b) To predict such clusters, we produce three cluster annotations for the same cells in panel a, with different granular cluster definitions (broad, median and fine). c) scTriangulate integration of Broad, Median and Fine clusters, using the default 2 TF-IDF scores (TF-IDF10 and TF-IDF5) or 1 TF-IDF score alone (TF-IDF10).

6. Our major concern is in presentation. Though some of us felt that the overall readability and presentation of the paper was satisfactory, we all agreed that the presentation and justification of the algorithm was extremely poor. We found the justification of the algorithm in terms of “cooperative game theory” to be confusing and misleading. In a manuscript where the main contribution is a new algorithm, the main text should contain a clear explanation of the algorithm in plain language. This explanation was missing, and details of how scTriangulate works was difficult to glean

from reading the figure captions, methods section, and supplements. Ultimately, we had to consult the source code and rederive the algorithm ourselves in order to understand it. Some of this would be improved by a reorganization of the supplement, though we also found notation errors in the supplement.

We thank the reviewer for the in-depth review of our manuscript and associated code. Based on the reviewer recommendations, we now provide clearer descriptions of the overall approach and justification of each of the steps of our approach in the Results along with more clear technical details and algorithm descriptions in the Methods. Where appropriate, we have simplified the Supplemental Figures, ensured they are called in a sequential manner and moved essential supplementary panels to the primary figures for improved clarity. Prior to resubmission, we reviewed the text with biologists lacking domain expertise in machine learning and statistics, to ensure all primary methods are clear, using plain language. We believe the new text significantly enhances readability and comprehension, while better emphasizing the statistical rigor of the approach.

7. However, our effort to rederive the algorithm led to some insights which might be useful in presenting and optimizing the algorithm. First, it seems like the Shapley value of an annotation (for a cell) for the four different quality scores (Reassign, TF-IDF(5) & (10), SCCAF) is a sum of the Shapley values for each score individually. It also seems like the Shapley of an annotation for a score depends only on its rank among the annotations for that cell. So, we suspect that the algorithm would be a lot simpler and faster if it pre-computed the Shapley value of each of the ranks $1, \dots, |A|$, (where $|A|$ is the number of annotations), put these values in a lookup table, and then quickly calculated the Shapley value of an annotation for a cell by summing the Shapley values associated with each of its ranks across the four quality scores. All this raises the question, that if the Shapley value is just a function of a rank, then why is it the best function, and not just the rank itself? Also, I wonder if each annotation could be given an overall quality score (like an Elo rating from chess) based on the per-cell comparisons, and then the annotations for each cell could be assigned based on this Elo rating.

We apologize for a lack of clarity regarding the algorithm descriptions in the main text and in the methods, which would be aided by simpler language along with the equations. To clarify the computation of the Shapley Value, first, a matrix of the stability metric scores for all players is input to a Value function (V) [Methods, Algorithm 1]. In this value function for Shapley analysis, rank is dynamically calculated for each combination of players given different stability metrics scores. Ranks from all stability metrics are derived independently for each coalition (group of players). As the reviewer notes, for each coalition ranks are computed for each stability metric, considering only the players present in that coalition and the ranks summed. A Shapley value is computed for the summed ranks derived in V for all coalitions for each player in the game. Hence, rather than the Shapley value being the sum of the Shapley values for each score individually, the Shapley value is derived from coalitional iteration of the stability metrics considering all players. In the Methods, the formal representation of the Value function is detailed in Algorithm 1 (note the text below in red has been revised):

Algorithm 1 scTriangulate Computing ($V(T \cup \{a\}) - V(T)$)

```
1: Input: annotation  $a$  and metric matrix  $D$ 
2:  $V=0$ 
3: for  $m \leftarrow \{Metric1, Metric2, Metric3, \dots\}$  do
4:     if  $\text{Rank}(D_{am} + 0.01) = |T \cup \{a\}|$  then
5:          $V \leftarrow V + |T \cup \{a\}|$ 
```

We now add additional clarifying text, similar to the above in the Methods (**Step3: Shapley Value determination and cell-cluster assignment**). Next, we would note that the rank computation is efficiently computed across all metrics in a vectorized manner [link 1] without verbose for loop, prior summarization ($V \leftarrow V + |T \cup \{a\}|$) [link 2]. Indeed, there is only one for loop in this function [link 3], which is for the iteration coalition. Hence, your recommended implementation is similar to what we are doing.

Link 1: <https://github.com/frankligy/scTriangulate/blob/main/sctriangulate/shapley.py#L104>

Link 2: <https://github.com/frankligy/scTriangulate/blob/main/sctriangulate/shapley.py#L110>

Link 3: <https://github.com/frankligy/scTriangulate/blob/main/sctriangulate/shapley.py#L98-L118>

However, the reviewer raises an important point about how the Shapley function compares to the use of using rank alone. We apologize for the confusion which we believe is due to the lack of explanation in our main text. First, we adopted Shapley because of its longstanding theoretical advantages in cooperative game theory. The goal of scTriangulate is to ensure that each single cell has the highest reassign, TFIDF and SCCAF score, however, this objective cannot be achieved by just adopting any single annotation, as the annotation with highest reassign score may have the lowest TF-IDF score and vice versa. To achieve this objective, our program cooperatively mixes-and-matches all considered solutions. Prior to implementation, we considered alternative solutions including competitive game theory. Different than cooperative game theory, in which annotations can be players, in competitive game theory, the players would be considered single-cells, the strategies would be annotations, and the payoff values would be the stability metrics, however, this design is not applicable for our approach, as it is very hard to compute Nash equilibrium in a N-player game (DOI: 10.1109/CSE.2009.17) compared to the classical prisoner dilemma or rock-scissor-paper (2 player). Furthermore, since we eventually need to assign a cell a cluster label (hard assignment versus soft assignment), each player (single cell) can not use a mixed strategy (e.g., 0.3 probability of choosing annotation1 and 0.7 probability of choosing annotation2), whereas Nash theorem doesn't guarantee each finite game will have a pure strategy Nash equilibrium. These two factors make the application of competitive game theory difficult.

Of note, a primary consideration for the use of cooperative game theory in scTriangulate is its focus on coalition. The computation of Shapley is effectively ranking (defined by a Value function), but the added benefit is that it exhausts every single possible coalition, measuring the marginal contribution of each player to ensure fair credit allocation, defined by three properties

(symmetry, dummy player and additivity axioms - see Osborne & Rubinstein textbook “A Course in Game theory” page 293). As shown in **Reviewer Figure 3** and detailed in **Reviewer 2 response 5**, we now systematically assess the contribution of different ranking methods in combination with different annotation importance approaches. As we note above, the `all_or_none` approach rank normalization approach (default option) only rewards a player if that cluster annotation has a higher score than all others for a given stability metric. This approach will potentially reduce noise introduced by retaining the source stability metric rankings. For example, if an annotation is just ranked 3rd or 4th, we have less confidence to conclude it is stable, in contrast to when a cluster is consistently ranked 1st. We first tested whether a simple rank-based strategy (excluding Shapley coalitional iteration), results in equivalent, improved or worse predictions to Shapley, considering different multimodal and unimodal datasets compared to Azimuth PBMC reference annotation assignments. This simple rank approach sums the score-derived ranks for each stability metric for each player for each cell, to obtain a final annotation rank for that player and cell. In addition to an absence of coalition iteration, this approach should produce more frequent “ties”, as different players will often have the same score (after considering the impact of the offset). Evaluation of simple ranking with `all_or_none` had the worst performance versus other parameter combinations in all benchmark datasets (**Reviewer Figure 3c**). These results suggest Shapley coalitional iteration exhausts the marginal contribution. We next evaluated the use of unadjusted ranks (aka classic), in place of `all_or_none`. For example, with unadjusted ranks, in an n player game, if a player is ranked 1st, it will be assigned a rank value of n , whereas the second top ranked will receive a rank value of $n-1$ point and so on. When evaluated using our benchmarking datasets, the simple rank-based annotation importance approach in conjunction with classic ranking, results in similar V-Measure predictions to Shapley for most datasets, but decreased performance for the dataset with the highest number of players (TEA-Seq). An example of when Shapley and simple rank based analyses will disagree, is in the case where there is a tie by simple rank, but a clear winner by Shapley. In the case of a tie, the cluster with fewest cells is determined the winner by default. However, for Shapley, since all possible coalitions are tested, cases where there is a statistical tie for a given stability metric will depend on which players are present and how “important” the annotation is given all possible combinations of players. No clear performance increases were achieved using Shapley with classic ranking. Hence, these data suggest that subtle differences between annotations are likely to be better captured with exhaustive iteration, using our default approach. In terms of speed, since rank-based method doesn’t involve an exhaustive iteration of all coalitions, it is generally faster, but the speed differences will be negligible when number of annotation (player) is less than 15, as computational complexity increase exponentially 2^{**n} for the shapley computation. Accordingly, we updated our codebase, tutorial and recommendations to include the `rank_based` method and recommend its use when the number of players (annotation sets from different sources) exceeds 15. We also provide a function called `salvage_run` to allow users to quickly test different options in the future.

We agree with the reviewer’s recommendation for the need to assess the stability/quality of the selected annotations. Hence, we now include the option to produce an average cluster Shapley value that serves as the analog of the Elo rating (<https://sctriangulate.readthedocs.io/en/latest/api.html#elo-rating-like>). This quality metric is

defined as the average of all cells' Shapley value in all clusters, normalized by the number of players, and further normalized by the number of clusters. Since the Shapley calculation is an additive process, the more players considered, the higher Shapley value will be. This quality metric can be used to approximate which cluster is more stable even for clusters derived from different datasets (the general objective of Elo rating in chess). Further, if a user analyzes dozens of possible annotation sources or clustering algorithms for a single dataset, all sources could in principle be ranked to assess their relative probability of being the "best". Hence, scTriangulate could be used to gauge which clustering algorithms generally are "the best players". We have added a description of this quality metric in the Methods (**Step3: Shapley Value determination and cell-cluster assignment**).

Additional comments and concerns:

8. Fig. 2: Small ADT-2 cluster in Fig. 2a isn't marked with ADT contribution in 2e, but instead has a high RNA contribution.

As an important point of clarification, while a cluster may be initially defined by ADTs (e.g., Leiden clustering in scanpy), it will be assessed for stability considering the union of all features from all evaluated modalities. As a result, the most distinguishing features may not be specific to the modality in which the cluster was originally defined. This is the case in the reviewer noted example in Figure 3e. Hence, an ADT-defined cluster may be best defined by RNAs, but should still be distinguished by ADTs. As ADTs are inherently less likely to be selected as top markers (only 31 were profiled in this analysis), it would be best to restrict marker inspection to a specific modality after performing scTriangulate (implemented via regular expression: https://github.com/frankligy/scTriangulate/blob/main/sctriangulate/main_class.py#L1814).

We now clarify this point in the initial algorithm description in two sections:

Results: *"To assess stability across all modalities, all molecular features are combined into a single object and jointly evaluated."*

Results: *"For multi-modality datasets, features from each modality are jointly considered (e.g., RNA, ATAC, ADT, splicing, mutation), with the contribution of each modality to each final cluster reported (Methods). Hence, while an initial cluster input in scTriangulate may have been defined by one modality (e.g., RNA), features from other modalities can support the stability of that population (e.g., ATAC). scTriangulate has no upper limit on modalities because it selects/highlights biological features from individual modalities in obtaining the final clustering solution."*

9. Some subfigures of Fig. 2 not referenced in the main text, lacking explanation of their relevance apart from the legend itself.

Thank you for pointing this out. We now explicitly clarify how modality specific features dominate distinct immune populations (ADT, ATAC, RNA) and further clarify as well as extend our benchmarking evaluations. Specifically:

Results: “For example, B-memory cells were resolved primarily by ATAC-Seq features, CD8 naive by ADT and B-naive by RNA (**Figure 3e**). In addition to these well-defined populations, scTriangulate identifies new clusters (CD4 and B-cell), frequently defined by gene accessibility or cell-surface proteins (**Figure 3f**). For example, we observe subdivision of CD4+ T-cells by the unique accessibility of a critical early T cell progenitor regulatory factor (ZBTB17 (Cheng et al. 2021)), which was not resolved at the level of gene-expression (**Supplementary Figure 3b**). These and other novel subdivisions, were further supported by cell surface protein expression in T-cells (CD45RA, CD95, KLRG1) (**Figure 3f**).”

10. No study of robustness of integration results to parameter choice (for e.g., default thresholds mentioned in line 112).

To communicate the newly presented evaluations described in **Reviewer 2 response 5** above, we have incorporated the below text:

Results: “Finally, to ensure the default parameters applied in this study are robust for diverse multimodal study designs, we compared them to alternative parameters, including alternative ranking strategies, cluster pruning options and decision strategies for annotation importance for all evaluated blood datasets. These evaluations demonstrate that scTriangulate defaults result in optimal multimodal clustering integration (**Supplementary Figure 7**). Hence, scTriangulate’s predictions are highly accurate relative to prior references while revealing reproducible novel populations.”

Discussion: “While the analytical settings applied in these studies were large programmatic defaults, it is likely these settings will not be optimal for all study designs. For example, the identification of rare bone marrow progenitor populations required the use of multiple TF-IDF scores to weight the selection of rare transitional cell-states with fewer unique expressed marker genes. Further, the integration of cluster predictions from dozens of sources (e.g., resolutions, modalities, reference annotations), will significantly increase runtime. For these reasons, the software includes intelligent defaults, alternative options for stability metric inclusion and flexible parameters (e.g., pruning threshold), that allow users to customize the outputs of scTriangulate out of the box. For example, the program will automatically substitute Shapley selection of final clusters with combined rank-based selection, when over 15 annotation sets are provided to reduce runtime. scTriangulate can be further customized beyond such options, to include new stability metrics or alternative rank-integration approaches.”

11. Label pruning and assignment to the nearest neighbor seems ad hoc and not a general consistent solution. In particular, it seems to break the philosophy of using the quality scores to assign annotations. Why not just default to the next highest scoring annotation for each cell?

This is an interesting question and to answer it we need to delve further into the rationale and decision making for pruning clusters. First, we assume that if a large initial cluster of cells exists, but only a small minority of cells with that annotation are considered to be "stable", that the overall predictions are more likely to be unreliable. For example, if a 1,000 cell cluster is initially annotated as Macrophages, but only 5% of the cells retain this annotation, the source annotation should not be considered reliable or specific to this small fraction of cells. For this reason, we recommend the default pruning threshold of 25%, to exclude such predictions and re-assign the label of these cells to the nearest neighbor, which is likely to represent the dominant annotation replacing the original Macrophage label. There are different hypotheses to why this might happen, assuming that our approach is perfect. First, it is assumed that scTriangulate retains this small proportion of cells because they have unique molecular features relative to their neighbors. Generally though, we expect that such cells are technical artifacts (e.g., high mitochondrial expression, dying cells) or that represent subtle biological differences (e.g., cell cycle phase). We often find such predictions supported by general visualization of mitochondrial or cell-cycle marker gene expression, following scTriangulate. If pruned cells represent technical outliers, assigning such cells to the next highest scoring annotation, will most often result in these still representing a small fraction of the second choice annotation, necessitating pruning. We reasoned that re-assigning these outlier cells to the best matching adjacent cluster was the most conservative solution after pruning. The concept of pruning and reclassification (rescue step) is used by various other tools, such as DeepST, where incorrectly assigned spots are reassigned (Long et al. 2022, bioRxiv) or low-quality matches in MARIO are filtered and reclassified (Zhu et al. 2021, bioRxiv). The specific method of reclassification is borrowed from tools such as ICGS2, which applies a NearestCentroid analysis. Here the centroids of stable clusters can be used as "landmarks" and cells belonging to unstable clusters will be reassigned to these centroids by nearest neighbors. " As demonstrated in our new evaluations of pruning thresholds, pruning with reclassification results in markedly improved correspondence to reference "silver-standard" annotations (Azimuth) (**Reviewer Figure 3d**). Choosing the correct threshold for pruning is non-obvious and should be explored by the user as now more clearly noted. As such, the "winning_proportion" can be visualized using the `plot_winners_statistics` function (<https://sctriangulate.readthedocs.io/en/latest/api.html#plot-winners-statistics>). By ranking the proportion of winning cells, we can assess how stable the source cluster is.

Additional rationale for pruning is described in the Methods (**Step 4: Results pruning and cell reclassification**), with benchmarking described in the **Results** and **Methods**.

12. Error in Fig. 4 legend - ...corresponds to source dataset annotations in panel b*

Thank you. We have made the correction.

13. Error in Fig 5. Legend d) Expression heatmap of final scTriangulate*

Thank you. We have made the correction.

14. Abbreviations such as GoT should be expanded (e.g., for those not familiar with clonal genotyping literature)

Thank you. We have now expanded this description to clarify the GoT protocol: *“Here, a small number of mutations were selectively detected from the same 10x Genomics scRNA-Seq MPN library, through sub-library amplification using the Genotyping of transcriptomes (GoT) protocol.”*

15. Exceedingly brief discussion of relevance in the field and future directions

Based on this reviewer feedback, we have substantially broadened our Discussion section, which now includes high-level conclusions from our study, noted statistical considerations, an overview of our new benchmarking, future opportunities and compatibility with new modalities/technologies, such as spatial transcriptomics. Please note, that spatial genomics analyses will likely require new types of stability metrics and optimization for different spatial platforms (e.g., low-resolution full-transcriptome, high-resolution reduced transcriptome) and niche-level considerations (e.g., BANKSY analysis) to obtain optimal integrations between distinct spatial and molecular modalities, hence, this is an important area for significant future development.

16. It would also be interesting to seeing how the method performs when provided with clusterings from a single modality at different resolutions (i.e., numbers of clusters). Will it always tend towards a higher or lower resolution or a combination thereof?

We now include an evaluation of multi-resolution integration of clusters in a unimodal PBMC dataset (**Supplementary Figure 6**). Here, benchmarking was performed for two different scTriangulate options: 1) two TF-IDF scores (default) and 2) one TF-IDF score (TF-IDF10 only). This analysis finds that while use of a single TF-IDF score has roughly equivalent performance to the use of a single Leiden resolution (resolution 1) based on comparison to PBMC reference cluster annotations (Azimuth) (**Supplementary Figure 6e**), the default analysis actually identifies many more granular populations (32 clusters with two TF-IDF scores, 28 with one TF-IDF score and 22 clusters with Leiden resolution 1). Such granular populations include prior-well defined CD14 monocyte subsets (classical, intermediate, and inflammatory) that are also observed in multiome and CITE-Seq datasets (**Supplementary Figures 5,6**). Hence, scTriangulate results will almost always produce a combination of different clustering resolutions, as demonstrated in **Supplementary Figure 6b**.

In the above example, when more clusters are obtained than observed in reference cell annotations, performance will inherently decrease. However, we argue that the Azimuth reference here is imperfect, as discussed throughout the paper, given that we identify numerous populations that correspond to those in the literature that are not present in the Azimuth. Hence, as a better test to assess the performance of scTriangulate given subclusters that weakly but consistently are associated with gene expression differences (i.e., cell-states), we produced a simulation dataset with Splatter. This simulation dataset explicitly tests our hypothesis that no

single-resolution is sufficient to optimally capture heterogeneity observed in a unimodal single-cell dataset using a verifiable ground-state truth (known transcriptionally distinct cell populations) (**Reviewer Figure 5**). Here, we hold out the simulation dataset ground-state truth cluster solution and test different solutions, each with under-clustering or over-clustering of different populations. While ensemble clustering fails to identify the correct solution, scTriangulate perfectly matches the simulation. Hence, we show that when different stable populations are present in different resolutions, scTriangulate can uniquely determine an optimal integrated solution.

Reviewer Figure 5. Simulated scRNA-Seq clusters are uniquely defined by scTriangulate.

a) As a ground-state truth to assess the specificity of predictions by scTriangulate, we simulated scRNA-Seq data for three highly distinct simulated populations (c1, c2, c3) and two subtly different subclusters (c4, c5) (Methods). b) To predict such clusters, we produce four cluster annotations for the same cells in panel a, with different granular cluster definitions (broad, median, fine and excessive). These include the noted under-clustered and over-clustered populations, which are not supported by gene expression differences. No individual resolution

intentionally mirrors ground-state truth. c) scTriangulate results using default parameters. The “resolutions” in which each cluster was selected from, is indicated in the cluster legend. d,e) Example ensemble clustering outputs from (d) HGPA and (e) MCLA applied to the four input resolutions. f) Benchmarking of the cluster assignments using three performance metrics (Homogeneity, Completeness, V-Measure) for three ensemble clustering techniques and scTriangulate. Green checkmark = matching cluster to simulated. Red X = non-matching cluster to simulated.

17. It's hard to understand how the scTriangulate actually works. The methods section is very unclear; variables seem to be describing multiple things, and understanding what Algorithm 1 (which computes the difference in team score within the overall Shapley Value calculation) was doing took several readings and examination of the python implementation on github.

As noted in Reviewer 2 response 6, we have now added additional clarifying descriptive language in our Methods and Results to increase comprehension, without the need to directly inspect the code.

18. The value provided by using the Shapley Value is not obvious. The ‘games’ being played for each cell seem to mostly be finding the highest ranking annotation across multiple metrics in a very complicated way – is this really all that different than just summing the ranks across all the quality scores and choosing the best sum? In particular, the actual collaborative element here is not clear to me.

Please see Reviewer 2 response 7.

19. The comparisons to alternate methods also need to be expanded. In figure 2f, where you compare scTriangulate’s agreement with a set of reference labels to that of other methods, the figure caption states that you are comparing with both Seurat WNN and single-modality Leiden clusters, but in the figure, there is only scTriangulate and different Seurat WNN resolutions. If by ‘single-modality Leiden clusters’ you mean those that were used in the scTriangulate pipeline, this is not made clear. Also, no justification is given for the choice of Seurat WNN clusterings as your only point of comparison (outside of the supplemental figures). I think the comparison against existing ensemble methods is important enough to be in the main paper instead of the supplement. Also, the metrics used for these comparisons (homogeneity, completeness, & v-measure) are neither explained or justified in this paper.

We understand the confusion. There was a caption error here, as we moved certain panels to the prior Supplementary Figure 3, without properly updating the main manuscript text. We now include all benchmarking for the TEA-Seq analysis, including TotalVI, ensembl and individual resolution clusterings, in the primary figure to avoid confusion. TotalVI and Seurat WNN were evaluated as alternative multimodal integration analysis methods, but in principle, could also be considered in the game. Indeed, these all demonstrate the unique ability of scTriangulate to

produce an optimal mix-and-match solution, relative to diverse algorithms and clustering resolutions. Details on the evaluation metrics are now provided in the methods section. We further now clarify the choice of metrics used for these comparisons (e.g., V-Measure), provide rationale for their use in the manuscript, and statistical benchmarking of these metrics relative to ARI and AMI. To clarify our choice of these metrics, we now include a detailed description of these evaluation metrics, along with rationale in the Methods (**scTriangulate Performance Evaluation > Performance metrics**).

We note that V-measure has been used in the evaluation of over 200 manuscripts for single-cell genomics data (Google Scholar). As an independent assessment, we compared Adjusted mutual information (AMI), Adjusted rand index (ARI) to V-measure, Homogeneity and Completeness in our TEA-Seq analyses as a frame of reference (**Reviewer Figure 6 and Figure 3g**). Note that AMI and V-measure produce near identical performance results (equivalent) and that ARI results, while lower, mirrors those observed for both V-measure and Completeness.

Reviewer Figure 6. Benchmarking of TEA-Seq scTriangulate multimodal prediction relative to diverse alternatives. Multiple measures of cluster agreement are shown for the same scTriangulate clustering solution relative to ensembl methods, individual modality-specific scanpy clustering resolutions and multimodal integration algorithms (multiple resolutions).

20. Confidence is misspelled in line 98

This sentence has been re-written.

21. The comma on line 235 is not necessary, and the sentence is overall quite clunky. Maybe it could be split into two sentences?

This has been corrected.

22. In equation (8) in the supplement, what is indexed by “i”? Should this be “a”? This same error is in the pseudo-code. This error is particularly bad because “i” is used to index cells immediately after Algorithm 1.

Thank you for identifying this. It has now been corrected as recommended.

23. What is the role of the 0.01 in the pseudo-code. Is this just a magic number to resolve tied ranks? If so, this seems arbitrary and not robust.

This is the offset which tests to see if two different players should have an equivalent rank for a specific stability metric. For example, if player 1 has a TF-IDF10 score of 0.912 and player 2 has a TF-IDF10 score of 0.904, they are assigned the same rank. We realize this notation was not clear in the Methods in Algorithm 1 and now provide details around the offset calculation in the text as described in more detail **Reviewer 2 response 5**.

24. I couldn't understand how the “Reassign” score was computed based on its description. Please clarify this.

The Reassign score evaluates the similarity of all cells to the collection of centroids in the dataset using the NearestCentroid function in scikit.learn. For this analysis, only the union of population discriminating marker genes are used (top 30, **Methods**). The empirical score produced from the software corresponds to a distance function, relative to the centroid as specified in the Methods equation 2. We now clarify this further in Methods:

Methods: “Reassign Score: The reassign score quantifies the reliability of a cell-to-cluster assignment when compared to its own centroid (nearest neighbor distance). We first defined the top-ranked 30 markers using an integrative strategy considering both log-fold change (effect size) and two-sided t-test p-value (significance) for each cluster versus all other cells, and pooled them together as the most discriminative features that can separate all clusters, the number of total pooled features is denoted by **L**. Next, a PCA analysis was performed on **L** to get reduced dimension **P**. Based on the discriminative features and the derived PCA space, we reduced the dimension of the dataset from **F** to **P** and the centroid of each examined cluster was derived based on these discriminative features, which arrives at a centroid matrix $\in \mathbf{R}^{P \times C}$ where **C** is the total number of clusters in each analyzed annotation set.”

REVIEWER COMMENTS

Reviewer #1 (Remarks to the Author):

We thank the authors for replying to our questions. Overall we are satisfied with the improvement of the paper.

Major comments

Supp Figure 8

- Using the MNP-Verse, the annotation does not align properly with the other atlas annotations and observations. It will be confusing to show that cDC1 is within the CD16+ mono population. Within the azimuth there is a score for how certain the cell is annotated. Could this potentially improve the annotation of the space?

Reviewer #2 (Remarks to the Author):

The revision fixes some of the major notation problems we identified, adds sensitive analyses for the various parameter settings, and has added some more clarifying text. It has also taken seriously our suggestion of replacing the Shapley value of an annotation with its rank and has compared these two approaches. And although I don't agree with all their conclusions based on their sensitivity analyses and these comparisons, they have nonetheless, addressed all our concerns except one: the presentation of the algorithm.

There seems to be no good reason to call this approach "game-theory". Specifically, because the Value function is so simple, the Shapley Value for an annotation is just a function of its rank. In fact, I bet with a bit of algebra, this equation can be rewritten without the summation over coalitions. This summation is why there is an exponential time complexity in number of annotations. This summation is also, as far as I can tell, the only justification for calling this technique "game-theory".

So, I am unconvinced by the justification of the approach in terms of cooperative games. Even if it were valid, I think that the game-theory terminology is unnecessarily confusing and provides little insight. I can't imagine anyone reading the new first paragraph of the section entitled "Step3: Shapley Value determination and cell-cluster assignment" and feeling like it added to their understanding of the algorithm.

Ultimately, authors are responsible for the content of their own manuscript. If they think that calling what they are doing "game-theory" is helpful to the reader, then who am I to disagree?

We again thank the reviewers for their consideration and helpful feedback. In this revision, we address the reviewer's two concerns with specific manuscript edits, a clarifying figure panel (Supplementary Figure 8b) and updates to the algorithm presentation. With these revisions, we believe we have fully addressed the reviewer concerns.

Reviewer #1 (Remarks to the Author):

We thank the authors for replying to our questions. Overall we are satisfied with the improvement of the paper.

Major comments

Supp Figure 8

- Using the MNP-Verse, the annotation does not align properly with the other atlas annotations and observations. It will be confusing to show that cDC1 is within the CD16+ mono population. Within the azimuth there is a score for how certain the cell is annotated. Could this potentially improve the annotation of the space?

As the reviewer notes, the clear division of cell populations in Wang et al. of MNP-Verse populations could represent technical issues that could be resolved through improved mappings. While we've already taken into account mapping quality in our pipeline to produce the best alignments we can (see below), the final results shown are subject to a number of variables: 1) mapping from single-cell to single-nucleus RNA-Seq, 2) the confidence of the MNP-Verse cell populations in healthy lung versus other organs used to derive these labels, 3) mapping quality and 4) visualization of projected clusters onto a prior-defined UMAP. The major issue we see with the MNP-Verse supervised classifications is a lack of definitive unique markers in the lung control samples with the Kim et al. reference used by Mulder et al. (PMID: 32385277). This is shown in **Reviewer Figure 1A** with predicted markers for each MNP-Verse cluster. Specifically, only 8 out of 18 MNP-Verse clusters had at least one unique marker (MarkerFinder Pearson $\rho > 0.4$). This may be due to the depth, quality or chemistry used by Kim et al., exclusion of non-normal tissues or non-lung, or simply poor markers for this high resolution of clusters. Importantly, we do not see clear markers in this Kim et al. data for cDC1 or related populations, which is an issue unrelated to mapping or scTriangulate. Hence, cDC1 mappings would be improved with more clearly markers in the lung control scRNA-Seq used in Mulder et al.

In terms of mapping, we applied the same multi-step alignment protocol used for the other independent lung atlas studies from Adams, Travaglini and Melms. Using this protocol, we first compute the top marker genes for each of the annotated cell clusters (MNP-Verse) for healthy lung samples in the reference dataset (**Reviewer Figure 1A**), output expression centroids for those cell clusters and markers and produced a preliminary cellHarmony classification against these reference centroids in the Wang et al. single-nucleus dataset. Because we are mapping across single-cell to single-nucleus data, these mappings will be sub-optimal (only 24.9% of the

Wang et al. cells map confidently to the MNP-Verse centroids). To improve these, we repeat this protocol in the mapped Wang et al. dataset, by re-deriving marker genes and centroids in the Wang et al. dataset with the confidently mapped MNP-Verse labels (24.9%) and re-classify all Wang et al. cells against these re-derived MNP-Verse clusters using Wang et al. reference cells. As the reviewer alluded to, this two-round mappings leverage quality score and will inherently be of higher confidence, as we are using matched single-nucleus reference centroids for the analysis. Since no Azimuth reference browser exists for the MNP-Verse, we also attempted to reproduce this Seurat-based workflow but obtained similar mapping issues (not shown). Again, these issues could be due to use of only normal Lung scRNA-Seq data from Kim et al., as we excluded cancer data and other tissues. Despite these issues, we do note that there is much better separation of Wang et al. cells for MNP-Verse clusters when we derive a UMAP based on MPN-Verse Wang et al. re-derived marker genes (**Reviewer Figure 1B, C**). We now replace **Supplementary Figure 8b** with these UMAPs.

Reviewer Figure 1. MNP-Verse Markers and Mappings. A) MarkerFinder heatmap of MNP-Verse marker genes, visualized in non-diseased lung scRNA-Seq (10x Chromium) from Kim et al. 2020. The top marker gene is displayed on the right next to each MNP-Verse annotated cluster (extracted from the author provided RDS file). Enriched gene-sets in each MNP-Verse cluster based on GO-Elite analysis (BioMarker Database in AltAnalyze). B) UMAP of MNP-Verse clusters produced through re-mapping of MNP-Verse clusters in the Wang et al. snRNA-Seq dataset. C) scTriangulate final clusters displayed in the UMAP from panel B.

Finally, while concerns exist in the normal lung annotations for cDC1 and related populations by Mulder et al., the scTriangulate cDC1 call replicates independent predictions from the LungMAP Cell Ref effort, Travaglini and Adams, validate that scTriangulate predictions are accurate both in terms of cluster stability and annotation label choice (**Figure 5d,f** and **Supplementary Figure 8A**). Hence, while the MNP-Verse mappings could be excluded entirely due to concerns with the specificity of markers in the reference dataset for some populations, we decided to include this evaluation dataset for completeness and transparency. We also note in the revised result section the technical difficulties mentioned above with respect to mapping scRNA to snRNA dataset, specifically:

“While in many cases, more granular cluster predictions are rejected by scTriangulate (e.g., ncMono and cMono in Adams) (**Supplementary Figure 8c**), we cannot rule out the possibility that such populations will be better resolved in larger patient cohorts, **by using a single-nucleus based reference, including more granular cell type predictions in the triangulation (e.g., MNP-Verse)**, or considering additional modalities.”

Reviewer #2 (Remarks to the Author):

The revision fixes some of the major notation problems we identified, adds sensitive analyses for the various parameter settings, and has added some more clarifying text. It has also taken seriously our suggestion of replacing the Shapley value of an annotation with its rank and has compared these two approaches. And although I don't agree with all their conclusions based on their sensitivity analyses and these comparisons, they have nonetheless, addressed all our concerns except one: the presentation of the algorithm.

There seems to be no good reason to call this approach "game-theory". Specifically, because the Value function is so simple, the Shapley Value for an annotation is just a function of its rank. In fact, I bet with a bit of algebra, this equation can be rewritten without the summation over coalitions. This summation is why there is an exponential time complexity in the number of annotations. This summation is also, as far as I can tell, the only justification for calling this technique "game-theory".

So, I am unconvinced by the justification of the approach in terms of cooperative games. Even if it were valid, I think that the game-theory terminology is unnecessarily confusing and provides

little insight. I can't imagine anyone reading the new first paragraph of the section entitled "Step3: Shapley Value determination and cell-cluster assignment" and feeling like it added to their understanding of the algorithm.

Ultimately, authors are responsible for the content of their own manuscript. If they think that calling what they are doing "game-theory" is helpful to the reader, then who am I to disagree?

We thank the reviewer for appreciating the novelty, importance, and performance of our approach to decisively integrate cell populations from diverse sources. The specific concern noted by the reviewer is that while Shapley is typically used to assign a payout to each player based on each of their individual contributions considering potentially complex variables, here we use Shapley to find the player with the greatest marginal contribution from stability metric ranks. Further, a more simple annotation importance approach (simple rank) recommended by the reviewer, has generally equivalent performance for one but not both of our evaluated ranking strategies (classic versus all-vs-none), suggesting that Shapley is unnecessarily complex and hence not efficient when considering dozens of annotations, whereas simple rank can work in some situations but not all (e.g., all-vs-none ranking). As Shapley works very well, we opted not to re-design our algorithm to evaluate possible mathematical alternatives, we revised the description of our approach to highlight "decision-level integration" in the title of the manuscript, abstract and methods. This title is inclusive of both Shapley and rank-based annotation importance options, both of which are available to users. We further refrain from drawing conceptual parallels to conventional cooperative game theory analyses and instead focus on a clear description of the algorithm. Please see highlighted edits in the manuscript.

REVIEWERS' COMMENTS

Reviewer #1 (Remarks to the Author):

The authors have addressed all our questions.